# Online Optimal Control with Linear Dynamics and Predictions: Algorithms and Regret Analysis

**Yingying Li**
SEAS
Harvard University
Cambridge, MA, 02138
yingyingli@g.harvard.edu

**Xin Chen**
SEAS
Harvard University
Cambridge, MA, 02138
chen_xin@g.harvard.edu

**Na Li**
SEAS
Harvard University
Cambridge, MA, 02138
nali@seas.harvard.edu

## Abstract

This paper studies the online optimal control problem with time-varying convex stage costs for a time-invariant linear dynamical system, where a finite lookahead window of accurate predictions of the stage costs are available at each time. We design online algorithms, Receding Horizon Gradient-based Control (RHGC), that utilize the predictions through finite steps of gradient computations. We study the algorithm performance measured by *dynamic regret*: the online performance minus the optimal performance in hindsight. It is shown that the dynamic regret of RHGC decays exponentially with the size of the lookahead window. In addition, we provide a fundamental limit of the dynamic regret for any online algorithms by considering linear quadratic tracking problems. The regret upper bound of one RHGC method almost reaches the fundamental limit, demonstrating the effectiveness of the algorithm. Finally, we numerically test our algorithms for both linear and nonlinear systems to show the effectiveness and generality of our RHGC.

## 1   Introduction

In this paper, we consider a $N$-horizon discrete-time sequential decision-making problem. At each time $t = 0, \ldots, N - 1$, the decision maker observes a state $x_t$ of a dynamical system, receives a $W$-step lookahead window of future cost functions of states and control actions, i.e. $f_t(x) + g_t(u), \ldots, f_{t+W-1}(x) + g_{t+W-1}(u)$, then decides the control input $u_t$ which drives the system to a new state $x_{t+1}$ following some known dynamics. For simplicity, we consider a linear time-invariant (LTI) system $x_{t+1} = Ax_t + Bu_t$ with $(A, B)$ known in advance. The goal is to minimize the overall cost over the $N$ time steps. This problem enjoys many applications in, e.g. data center management [1, 2], robotics [3], autonomous driving [4, 5], energy systems [6], manufacturing [7, 8]. Hence, there has been a growing interest on the problem, from both control and online optimization communities.

In the control community, studies on the above problem focus on economic model predictive control (EMPC), which is a variant of model predictive control (MPC) with a primary goal on optimizing economic costs [9, 10, 11, 12, 13, 14, 15, 16]. Recent years have seen a lot of attention on the optimality performance analysis of EMPC, under both time-invariant costs [17, 18, 19] and time-varying costs [20, 12, 14, 21, 22]. However, most studies focus on asymptotic performance and there is still limited understanding on the non-asymptotic performance, especially under time-varying costs. Moreover, for computationally efficient algorithms, e.g. suboptimal MPC and inexact MPC [23, 24, 25, 26], there is limited work on the optimality performance guarantee.

In online optimization, on the contrary, there are many papers on the non-asymptotic performance analysis, where the performance is usually measured by regret, e.g., static regrets[27, 28], dynamic regrets[29], etc., but most work does not consider predictions and/or dynamical systems. Further, motivated by the applications with predictions, e.g. predictions of electricity prices in data center

management problems [30, 31], there is a growing interest on the effect of predictions on the online problems [32, 33, 30, 34, 31, 35, 36]. However, though some papers consider switching costs which can be viewed as a simple and special dynamical model [37, 36], there is a lack of study on the general dynamical systems and on how predictions affect the online problem with dynamical systems.

In this paper, we propose novel gradient-based online control algorithms, receding horizon gradient-based control (RHGC), and provide nonasymptotic optimality guarantees by dynamic regrets. RHGC can be based on many gradient methods, e.g. vanilla gradient descent, Nesterov's accelerated gradient, triple momentum, etc., [38, 39]. Due to the space limit, this paper only presents receding horizon gradient descent (RHGD) and receding horizon triple momentum (RHTM). For the theoretical analysis, we assume strongly convex and smooth cost functions, whereas applying RHGC does not require these conditions. Specifically, we show that the regret bounds of RHGD and RHTM decay exponentially with the prediction window's size $W$, demonstrating that our algorithms efficiently utilize the prediction. Besides, our regret bounds decrease when the system is more "agile" in the sense of a controllability index [40]. Further, we provide a fundamental limit for any online control algorithms and show that the fundamental lower bound almost matches the regret upper bound of RHTM. This indicates that RHTM achieves near-optimal performance at least in the worst case. We also provide some discussion on the classic linear quadratic tracking problems, a widely studied control problem in literature, to provide more insightful interpretations of our results. Finally, we numerically test our algorithms. In addition to linear systems, we also apply RHGC to a nonlinear dynamical system: path tracking by a two-wheeled robot. Results show that RHGC works effectively for nonlinear systems though RHGC is only presented and theoretical analyzed on LTI systems.

Results in this paper are built on a paper on online optimization with switching costs [36]. Compared with [36], this paper studies online optimal control with *general linear dynamics*, which includes [36] as a special case; and studies how the system controllability index affects the regrets.

There has been some recent work on online optimal control problems with time-varying costs [41, 42, 37, 43] and/or time-varying disturbances [43], but most papers focus on the no-prediction cases. As we show later in this paper, these algorithms can be used in our RHGC methods as initialization oracles. Moreover, our regret analysis shows that RHGC can reduce the regret of these no-prediction online algorithms by a factor exponentially decaying with the prediction window's size.

Finally, we would like to mention another related line of work: learning-based control [44, 45, 46, 47, 48]. In some sense, the results in this paper are orthogonal to that of the learning-based control, because the learning-based control usually considers a time-invariant environment but unknown dynamics, and aims to learn system dynamics or optimal controllers by data; while this paper considers a time-varying scenario with known dynamics but changing objectives and studies decision making with limited predictions. It is an interesting future direction to combine the two lines of work for designing more applicable algorithms.

**Notations.** Consider matrices $A$ and $B$, $A \geq B$ means $A - B$ is positive semidefinite and $[A, B]$ denotes a block matrix. The norm $\| \cdot \|$ refers to the $L_2$ norm for both vectors and matrices. Let $x^i$ denote the $i$th entry of the vector. Consider a set $\mathcal{I} = \{k_1, \ldots, k_m\}$, then $x^{\mathcal{I}} = (x^{k_1}, \ldots, x^{k_m})^\top$, and $A(\mathcal{I}, :)$ denotes the $\mathcal{I}$ rows of matrix $A$ stacked together. Let $I_m$ be an identity matrix in $\mathbb{R}^{m \times m}$.

## 2 Problem formulation and preliminaries

Consider a finite-horizon discrete-time optimal control problem with time-varying cost functions $f_t(x_t) + g_t(u_t)$ and a linear time-invariant (LTI) dynamical system:

$$\min_{\mathbf{x}, \mathbf{u}} \quad J(\mathbf{x}, \mathbf{u}) = \sum_{t=0}^{N-1} [f_t(x_t) + g_t(u_t)] + f_N(x_N) \tag{1}$$

$$\text{s.t.} \quad x_{t+1} = A x_t + B u_t, \quad t \geq 0$$

where $x_t \in \mathbb{R}^n$, $u_t \in \mathbb{R}^m$, $\mathbf{x} = (x_1^\top, \ldots, x_N^\top)^\top$, $\mathbf{u} = (u_0^\top, \ldots, u_{N-1}^\top)^\top$, $x_0$ is given, $f_N(x_N)$ is the terminal cost.[1] To solve the optimal control problem (1), all cost functions from $t = 0$ to $t = N$ are needed. However, at each time $t$, usually only a finite lookahead window of cost functions are available and the decision maker needs to make an online decision $u_t$ using the available information.

In particular, we consider a simplified prediction model: at each time $t$, the decision maker obtains accurate predictions for the next $W$ time steps, $f_t, g_t, \ldots, f_{t+W-1}, g_{t+W-1}$, but no further prediction beyond these $W$ steps, meaning that $f_{t+W}, g_{t+W}, \ldots$ can even be adversarially generated. Though this prediction model may be too optimistic in the short term and over pessimistic in the long term, this model i) captures a commonly observed phenomenon in predictions that short-term predictions are usually much more accurate than the long-term predictions; ii) allows researchers to derive insights for the role of predictions and possibly to extend to more complicated cases [31, 30, 49, 50].

The online optimal control problem is described as follows: at each time step $t = 0, 1, \ldots,$

- the agent observes state $x_t$ and receives prediction $f_t, g_t, \ldots, f_{t+W-1}, g_{t+W-1}$;
- the agent decides and implements a control $u_t$ and suffers the cost $f_t(x_t) + g_t(u_t)$;
- the system evolves to the next state $x_{t+1} = Ax_t + Bu_t$.[2]

An online control algorithm, denoted as $\mathcal{A}$, can be defined as a mapping from the prediction information and the history information to the control action, denoted by $u_t(\mathcal{A})$:

$$u_t(\mathcal{A}) = \mathcal{A}(x_t(\mathcal{A}), \ldots, x_0(\mathcal{A}), \{f_s, g_s\}_{s=0}^{t+W-1}), \quad t \geq 0, \tag{2}$$

where $x_t(\mathcal{A})$ is the state generated by implementing $\mathcal{A}$ and $x_0(\mathcal{A}) = x_0$ is given.

This paper evaluates the performance of online control algorithms by comparing against the optimal control cost $J^*$ in hindsight, that is, $J^* := \min\{J(\mathbf{x}, \mathbf{u}) \mid x_{t+1} = Ax_t + Bu_t, \ \forall t \geq 0\}$.

In this paper, the performance of an online algorithm $\mathcal{A}$ is measured by [3]

$$\text{Regret}(\mathcal{A}) := J(\mathcal{A}) - J^* = J(\mathbf{x}(\mathcal{A}), \mathbf{u}(\mathcal{A})) - J^*, \tag{3}$$

which is sometimes called as *dynamic regret* [29, 51] or *competitive difference* [52]. Another popular regret notion is the static regret, which compares the online performance with the optimal static controller/policy [42, 41]. The benchmark in static regret is weaker than that in dynamic regret because the optimal controller may be far from being static, and it has been shown in literature that $o(N)$ static regret can be achieved even without predictions (i.e., $W = 0$). Thus, we will focus on the dynamic regret analysis and study how predictions can improve the dynamic regret.

**Example 1** (Linear quadratic (LQ) tracking). *Consider a discrete-time tracking problem for a system $x_{t+1} = Ax_t + Bu_t$. The goal is to minimize the quadratic tracking loss of a trajectory $\{\theta_t\}_{t=0}^N$*

$$J(\mathbf{x}, \mathbf{u}) = \frac{1}{2} \sum_{t=0}^{N-1} \left[ (x_t - \theta_t)^\top Q_t (x_t - \theta_t) + u_t^\top R_t u_t \right] + \frac{1}{2}(x_N - \theta_N)^\top Q_N (x_N - \theta_N).$$

*In practice, it is usually difficult to know the complete trajectory $\{\theta_t\}_{t=0}^N$ a priori, what are revealed are usually the next few steps, making it an online control problem with predictions.*

**Assumptions and useful concepts.** Firstly, we assume controllability, which is standard in control theory and roughly means that the system can be steered to any state by proper control inputs [53].

**Assumption 1.** *The LTI system $x_{t+1} = Ax_t + Bu_t$ is controllable.*

It is well-known that any controllable LTI system can be linearly transformed to a canonical form [40] and the linear transformation can be computed efficiently a priori using $A$ and $B$, which can further be used to reformulate the cost functions $f_t, g_t$. Thus, without loss of generality, this paper only considers LTI systems in the canonical form, defined as follows.

**Definition 1** (Canonical form). *A system $x_{t+1} = Ax_t + Bu_t$ is said to be in the canonical form if*

___

[2]We assume known $A, B$, no process noises, state feedback, and leave relaxing assumptions as future work.
[3]The optimality gap depends on initial state $x_0$ and $\{f_t, g_t\}_{t=0}^N$, but we omit them for simplicity of notation.

*where each \* represents a (possibly) nonzero entry, and the rows of $B$ with $1$ are the same rows of $A$ with \* and the indices of these rows are denoted as $\{k_1, \ldots, k_m\} =: \mathcal{I}$. Moreover, let $p_i = k_i - k_{i-1}$ for $1 \leq i \leq m$, where $k_0 = 0$. The* controllability index *of a canonical-form $(A, B)$ is defined as*

$$p = \max\{p_1, \ldots, p_m\}.$$

Next, we introduce assumptions on the cost functions and their optimal solutions.

**Assumption 2.** *Assume $f_t$ is $\mu_f$ strongly convex and $l_f$ Lipschitz smooth for $0 \leq t \leq N$, and $g_t$ is convex and $l_g$ Lipschitz smooth for $0 \leq t \leq N - 1$ for some $\mu_f, l_f, l_g > 0$.*

**Assumption 3.** *Assume the minimizers to $f_t, g_t$, denoted as $\theta_t = \arg\min_x f_t(x)$, $\xi_t = \arg\min_u g_t(u)$, are uniformly bounded, i.e. there exist $\bar{\theta}, \bar{\xi}$ such that $\|\theta_t\| \leq \bar{\theta}$, $\|\xi_t\| \leq \bar{\xi}$, $\forall t$.*

These assumptions are commonly adopted in convex analysis. The uniform bounds rule out extreme cases. Notice that the LQ tracking problem in Example 1 satisfies Assumption 2 and 3 if $Q_t, R_t$ are positive definite with uniform bounds on the eigenvalues and if $\theta_t$ are uniformly bounded for all $t$.

## 3 Online control algorithms: receding horizon gradient-based control

This section introduces our online control algorithms, receding horizon gradient-based control (RHGC). The design is by first converting the online control problem to an equivalent online optimization problem with *finite temporal-coupling* costs, then designing gradient-based online optimization algorithms by utilizing this finite temporal-coupling property.

### 3.1 Problem transformation

Firstly, we notice that the offline optimal control problem (1) can be viewed as an optimization with equality constraints over $\mathbf{x}$ and $\mathbf{u}$. The individual stage cost $f_t(x_t) + g_t(u_t)$ only depends on the current $x_t$ and $u_t$ but the equality constraints couple $x_t$, $u_t$ with $x_{t+1}$ for each $t$. In the following, we will rewrite (1) in an equivalent form of an *unconstrained* optimization problem on some entries of $x_t$ for all $t$, but the new stage cost at each time $t$ will depend on these new entries across a few nearby time steps. We will harness this structure to design our online algorithm.

In particular, the entries of $x_t$ adopted in the reformulation are: $x_t^{k_1}, \ldots, x_t^{k_m}$, where $\mathcal{I} = \{k_1, \ldots, k_m\}$ is defined in Definition 1. For ease of notation, we define

$$z_t := (x_t^{k_1}, \ldots, x_t^{k_m})^\top, \quad t \geq 0 \tag{4}$$

and write $z_t^j = x_t^{k_j}$ where $j = 1, \ldots, m$. Let $\mathbf{z} := (z_1^\top, \ldots, \ldots, z_N^\top)^\top$. By the canonical-form equality constraint $x_t = A x_{t-1} + B u_{t-1}$, we have $x_t^i = x_{t-1}^{i+1}$ for $i \notin \mathcal{I}$, so $x_t$ can be represented by $z_{t-p+1}, \ldots, z_t$ in the following way:

$$x_t = (\underbrace{z_{t-p_1+1}^1, \ldots, z_t^1}_{p_1}, \underbrace{z_{t-p_2+1}^2, \ldots, z_t^2}_{p_2}, \ldots, \underbrace{z_{t-p_m+1}^m, \ldots, z_t^m}_{p_m})^\top, \quad t \geq 0, \tag{5}$$

where $z_t$ for $t \leq 0$ is determined by $x_0$ in a way to let (5) hold for $t = 0$. For ease of exposition and without loss of generality, we consider $x_0 = 0$ in this paper; then we have $z_t = 0$ for $t \leq 0$. Similarly, $u_t$ can be determined by $z_{t-p+1}, \ldots, z_t, z_{t+1}$ by

$$u_t = z_{t+1} - A(\mathcal{I}, :)x_t = z_{t+1} - A(\mathcal{I}, :)(z_{t-p_1+1}^1, \ldots, z_t^1, \ldots, z_{t-p_m+1}^m, \ldots, z_t^m)^\top, \quad t \geq 0 \tag{6}$$

where $A(\mathcal{I}, :)$ consists of $k_1, \ldots, k_m$ rows of $A$.

It is straightforward to verify that equations (4, 5, 6) describe a bijective transformation between $\{(\mathbf{x}, \mathbf{u}) \mid x_{t+1} = A x_t + B u_t\}$ and $\mathbf{z} \in \mathbb{R}^{mN}$, since the LTI constraint $x_{t+1} = A x_t + B u_t$ is naturally embedded in the relation (5, 6). Therefore, based on the transformation, an optimization problem with respect to $\mathbf{z} \in \mathbb{R}^{mN}$ can be designed to be equivalent with (1). Notice that the resulting optimization problem has no constraint on $\mathbf{z}$. Moreover, the cost functions on $\mathbf{z}$ can be obtained by substituting (5, 6) into $f_t(x_t)$ and $g_t(u_t)$, i.e. $\tilde{f}_t(z_{t-p+1}, \ldots, z_t) := f_t(x_t)$ and $\tilde{g}_t(z_{t-p+1}, \ldots, z_t, z_{t+1}) := g_t(u_t)$. Correspondingly, the objective function of the equivalent optimization with respect to $\mathbf{z}$ is

$$C(\mathbf{z}) := \sum_{t=0}^{N} \tilde{f}_t(z_{t-p+1}, \ldots, z_t) + \sum_{t=0}^{N-1} \tilde{g}_t(z_{t-p+1}, \ldots, z_{t+1}) \tag{7}$$

$C(\mathbf{z})$ has many nice properties, some of which are formally stated below.

**Lemma 1.** *The function $C(\mathbf{z})$ has the following properties:*

    *i)* $C(\mathbf{z})$ *is* $\mu_c = \mu_f$ *strongly convex and* $l_c$ *smooth for* $l_c = pl_f + (p+1)l_g\|[I_m, -A(\mathcal{I},:)]\|^2$.

    *ii)* *For any* $(\mathbf{x}, \mathbf{u})$ *s.t.* $x_{t+1} = Ax_t + Bu_t$, $C(\mathbf{z}) = J(\mathbf{x}, \mathbf{u})$ *where* $\mathbf{z}$ *is defined in (4). Conversely,* $\forall\, \mathbf{z}$, *the* $(\mathbf{x}, \mathbf{u})$ *determined by (5,6) satisfies* $x_{t+1} = Ax_t + Bu_t$ *and* $J(\mathbf{x}, \mathbf{u}) = C(\mathbf{z})$;

    *iii)* *Each stage cost* $\tilde{f}_t + \tilde{g}_t$ *in (7) only depends on* $z_{t-p+1}, \ldots, z_{t+1}$.

Property ii) implies that any online algorithm for deciding $\mathbf{z}$ can be translated to an online algorithm for $\mathbf{x}$ and $\mathbf{u}$ by (5, 6) with the same costs. Property iii) highlights one nice property, finite temporal-coupling, of $C(\mathbf{z})$, which serves as a foundation for our online algorithm design.

**Example 2.** *For illustration, consider the following dynamical system with $n = 2$, $m = 1$:*

$$\begin{bmatrix} x_{t+1}^1 \\ x_{t+1}^2 \end{bmatrix} = \begin{bmatrix} 0 & 1 \\ a_1 & a_2 \end{bmatrix} \begin{bmatrix} x_t^1 \\ x_t^2 \end{bmatrix} + \begin{bmatrix} 0 \\ 1 \end{bmatrix} u_t \tag{8}$$

*Here,* $k_1 = 2$, $\mathcal{I} = \{2\}$, $A(\mathcal{I},:) = (a_1, a_2)$, *and* $z_t = x_t^2$. *By (8),* $x_t^1 = x_{t-1}^2$ *and* $x_t = (z_{t-1}, z_t)^\top$. *Similarly,* $u_t = x_{t+1}^2 - A(\mathcal{I},:)x_t = z_{t+1} - A(\mathcal{I},:)(z_{t-1}, z_t)^\top$. *Hence,* $\tilde{f}_t(z_{t-1}, z_t) = f_t(x_t) = f_t((z_{t-1}, z_t)^\top)$, $\tilde{g}_t(z_{t-1}, z_t, z_{t+1}) = g_t(u_t) = g_t(z_{t+1} - A(\mathcal{I},:)(z_{t-1}, z_t)^\top)$.

*Remark* 1. This paper considers a reparameterization method with respect to states $\mathbf{x}$ via the canonical form, and it might be interesting to compare it with the more direct reparameterization with respect to control inputs $\mathbf{u}$. The control-based reparameterization has been discussed in literature [54]. It has been observed in [54] that when $A$ is not stable, the condition number of the cost function derived from the control-based reparameterization goes to infinity as $W \to +\infty$, which may result in computation issues when $W$ is large. However, the state-based reparameterization considered in this paper can guarantee bounded condition number for all $W$ even for unstable $A$, as shown in Lemma 1. This is one major advantage of the state-based reparameterization method considered in this paper.

## 3.2 Online algorithm design: RHGC

This section introduces our RHGC based on the reformulation (7) and inspired by [36]. As mentioned earlier, any online algorithm for $z_t$ can be translated to an online algorithm for $x_t, u_t$. Hence, we will focus on designing an online algorithm for $z_t$ in the following. By the finite temporal-coupling property of $C(\mathbf{z})$, the partial gradient of the *total cost* $C(\mathbf{z})$ only depends on the finite neighboring stage costs $\{\tilde{f}_\tau, \tilde{g}_\tau\}_{\tau=t}^{t+p-1}$ and finite neighboring stage variables $(z_{t-p}, \ldots, z_{t+p}) =: z_{t-p:t+p}$.

$$\frac{\partial C}{\partial z_t}(\mathbf{z}) = \sum_{\tau=t}^{t+p-1} \frac{\partial \tilde{f}_\tau}{\partial z_t}(z_{\tau-p+1}, \ldots, z_\tau) + \sum_{\tau=t-1}^{t+p-1} \frac{\partial \tilde{g}_\tau}{\partial z_t}(z_{\tau-p+1}, \ldots, z_{\tau+1})$$

Without causing any confusion, we use $\frac{\partial C}{\partial z_t}(z_{t-p:t+p})$ to denote $\frac{\partial C}{\partial z_t}(\mathbf{z})$ for highlighting the local dependence. Thanks to the local dependence, despite the fact that not all the future costs are available, it is still possible to compute the partial gradient of the total cost by using only a finite lookahead window of the cost functions. This observation motivates the design of our receding horizon gradient-based control (RHGC) methods, which are the online implementation of gradient methods, such as vanilla gradient descent, Nesterov's accelerated gradient, triple momentum, etc., [38, 39].

---

**Algorithm 1:** Receding Horizon Gradient Descent (RHGD)

---

1: **inputs:** Canonical form $(A, B)$, $W \geq 1$, $K = \lfloor \frac{W-1}{p} \rfloor$, stepsize $\gamma_g$, initialization oracle $\varphi$.
2: **for** $t = 1 - W : N - 1$ **do**
3:    *Step 1:* initialize $z_{t+W}(0)$ by oracle $\varphi$.
4:    **for** $j = 1, \ldots, K$ **do**
5:       *Step 2:* update $z_{t+W-jp}(j)$ by gradient descent
        $z_{t+W-jp}(j) = z_{t+W-jp}(j-1) - \gamma_g \frac{\partial C}{\partial z_{t+W-jp}}(z_{t+W-(j+1)p:t+W-(j-1)p}(j-1))$.
6:    **end for**
7:    *Step 3:* compute $u_t$ by $z_{t+1}(K)$ and the observed state $x_t$: $u_t = z_{t+1}(K) - A(\mathcal{I},:)x_t$
8: **end for**

---

Firstly, we illustrate the main idea of RHGC by receding horizon gradient descent (RHGD) based on vanilla gradient descent. In RHGD (Algorithm 1), index $j$ refers to the iteration number of the

corresponding gradient update of $C(\mathbf{z})$. There are two major steps to decide $z_t$. Step 1 is initializing the decision variables $\mathbf{z}(0)$. Here, we do not restrict the initialization algorithm $\varphi$ and allow any oracle/online algorithm without using lookahead information, i.e. $z_{t+W}(0)$ is selected based only on the information up to $t + W - 1$: $z_{t+W}(0) = \varphi(\{\tilde{f}_s, \tilde{g}_s\}_{s=0}^{t+W-1})$. One example of $\varphi$ will be provided in Section 4. Step 2 is using the $W$-lookahead costs to conduct gradient updates. Notice that the gradient update from $z_\tau(j-1)$ to $z_\tau(j)$ is implemented in a backward order of $\tau$, i.e. from $\tau = t + W$ to $\tau = t$. Moreover, since the partial gradient $\frac{\partial C}{\partial z_t}$ requires the local decision variables $z_{t-p:t+p}$, given $W$-lookahead information, RHGD can only conduct $K = \lfloor \frac{W-1}{p} \rfloor$ iterations of gradient descent for the total cost $C(\mathbf{z})$. For more discussion, we refer the reader to [36] for the $p = 1$ case.

In addition to RHGD, RHGC can also incorporate accelerated gradient methods in the same way, such as Nesterov's accelerated gradient and triple momentum. For the space limit, we only formally present receding horizon triple momentum (RHTM) in Algorithm 2 based on triple momentum [39]. RHTM also consists of two major steps when determining $z_t$: initialization and gradient updates based on the lookahead window. The two major differences from RHGD are that the decision variables in RHTM include not only $z(j)$ but also auxiliary variables $\omega(j)$ and $y(j)$, which are adopted in triple momentum to accelerate the convergence, and that the gradient update is by triple momentum instead of gradient descent. Nevertheless, RHTM can also conduct $K = \lfloor \frac{W-1}{p} \rfloor$ iterations of triple momentum for $C(\mathbf{z})$ since the triple momentum update requires the same neighboring cost functions.

Though it appears that RHTM does not fully exploit the lookahead information since only a few gradient updates are used, in Section 5, we show that RHTM achieves near-optimal performance with respect to $W$, which means that RHTM successfully extracts and utilizes the prediction information.

Finally, we briefly introduce MPC[55] and suboptimal MPC[23], and compare them with our algorithms. MPC tries to solve a $W$-stage optimization at each $t$ and implements the first control input. Suboptimal MPC, as a variant of MPC aiming at reducing computation, conducts an optimization method only for a few iterations without solving the optimization completely. Our algorithm's computation time is similar to that of suboptimal MPC with a few gradient iterations. However, the major difference between our algorithm and suboptimal MPC is that suboptimal MPC conducts gradient updates for a truncated $W$-stage optimal control problem based on $W$-lookahead information, while our algorithm is able to conduct gradient updates for the complete $N$-stage optimal control problem based on the same $W$-lookahead information by utilizing the reformulation (4, 5, 6, 7).

## 4  Regret upper bounds

Because our RHTM (RHGD) is designed to exactly implement the triple momentum (gradient descent) of $C(\mathbf{z})$ for $K$ iterations, it is straightforward to have the following regret guarantees that connect the regrets of RHTM and RHGD with the regret of the initialization oracle $\varphi$,

---

**Algorithm 2:** Receding Horizon Triple Momentum (RHTM)

---

   **inputs:** Canonical form $(A, B)$, $W \geq 1$, $K = \lfloor \frac{W-1}{p} \rfloor$, stepsizes $\gamma_c, \gamma_z, \gamma_\omega, \gamma_y > 0$, oracle $\varphi$.
   **for** $t = 1 - W : N - 1$ **do**
      *Step 1:* initialize $z_{t+W}(0)$ by oracle $\varphi$, then let $\omega_{t+W}(-1), \omega_{t+W}(0), y_{t+W}(0)$ be $z_{t+W}(0)$
      **for** $j = 1, \ldots, K$ **do**
         *Step 2:* update $\omega_{t+W-jp}(j), y_{t+W-jp}(j), z_{t+W-jp}(j)$ by triple momentum.

$$\omega_{t+W-jp}(j) = (1 + \gamma_\omega)\omega_{t+W-jp}(j-1) - \gamma_\omega \omega_{t+W-jp}(j-2)$$
$$- \gamma_c \frac{\partial C}{\partial y_{t+W-jp}}(y_{t+W-(j+1)p:t+W-(j-1)p}(j-1))$$
$$y_{t+W-jp}(j) = (1 + \gamma_y)\omega_{t+W-jp}(j) - \gamma_y \omega_{t+W-jp}(j-1)$$
$$z_{t+W-jp}(j) = (1 + \gamma_z)\omega_{t+W-jp}(j) - \gamma_z \omega_{t+W-jp}(j-1)$$

   **end for**
      *Step 3:* compute $u_t$ by $z_{t+1}(K)$ and the observed state $x_t$: $u_t = z_{t+1}(K) - A(\mathcal{I}, :)x_t$
   **end for**

---

**Theorem 1.** *Consider $W \geq 1$ and stepsizes $\gamma_g = \frac{1}{l_c}$, $\gamma_c = \frac{1+\phi}{l_c}$, $\gamma_\omega = \frac{\phi^2}{2-\phi}$, $\gamma_y = \frac{\phi^2}{(1+\phi)(2-\phi)}$, $\gamma_z = \frac{\phi^2}{1-\phi^2}$, $\phi = 1 - 1/\sqrt{\zeta}$, and let $\zeta = l_c/\mu_c$ denote $C(\mathbf{z})$'s condition number. For any oracle $\varphi$,*

$$\text{Regret}(RHGD) \leq \zeta \left( \frac{\zeta - 1}{\zeta} \right)^K \text{Regret}(\varphi), \quad \text{Regret}(RHTM) \leq \zeta^2 \left( \frac{\sqrt{\zeta} - 1}{\sqrt{\zeta}} \right)^{2K} \text{Regret}(\varphi)$$

*where $K = \lfloor \frac{W-1}{p} \rfloor$, $\text{Regret}(\varphi)$ is the regret of the initial controller: $u_t(0) = z_{t+1}(0) - A(\mathcal{I},:)x_t(0)$.*

Theorem 1 suggests that for any online algorithm $\varphi$ without predictions, RHGD and RHTM can use predictions to lower the regret by a factor of $\zeta(\frac{\zeta-1}{\zeta})^K$ and $\zeta^2(\frac{\sqrt{\zeta}-1}{\sqrt{\zeta}})^{2K}$ respectively via additional $K = \lfloor \frac{W-1}{p} \rfloor$ gradient updates. Moreover, the factors decay exponentially with $K = \lfloor \frac{W-1}{p} \rfloor$, and $K$ almost linearly increases with $W$. This indicates that RHGD and RHTM improve the performance exponentially fast with an increase in the prediction window $W$ for any initialization method. In addition, $K = \lfloor \frac{W-1}{p} \rfloor$ decreases with $p$, implying that the regrets increase with the controllability index $p$ (Definition 1). This is intuitive because $p$ roughly indicates how fast the controller can influence the system state effectively: the larger the $p$ is, the longer it takes. To see this, consider Example 2. Since $u_{t-1}$ does not directly affect $x_t^1$, it takes at least $p = 2$ steps to change $x_t^1$ to a desirable value. Finally, RHTM's regret decays faster than RHGD's, which is intuitive because triple momentum converges faster than gradient descent. Thus, we will focus on RHTM in the following.

**An initialization method: follow the optimal steady state (FOSS).** To complete the regret analysis for RHTM, we provide a simple initialization method, FOSS, and its dynamic regret bound. As mentioned before, any online control algorithm without predictions, e.g. [42, 41], can be applied as an initialization oracle $\varphi$. However, most literature study static regrets rather than dynamic regrets.

**Definition 2** (Follow the optimal steady state (FOSS)). *The optimal steady state for stage cost $f(x) + g(u)$ refers to $(x^e, u^e) := \arg\min_{x=Ax+Bu}(f(x) + g(u))$.*

*Follow the optimal steady state algorithm (FOSS) first solves the optimal steady state $(x_t^e, u_t^e)$ for cost $f_t(x) + g_t(u)$, then determines $z_{t+1}$ by $x_t^e$, i.e. $z_{t+1} = (x_t^{e,k_1}, \ldots, x_t^{e,k_m})^\top$ at each $t + 1$.*

FOSS is motivated by the fact that the optimal steady state cost is the optimal infinite-horizon average cost for LTI systems with time-invariant cost functions [56], so FOSS should yield acceptable performance at least for slowly changing cost functions. Nevertheless, we admit that FOSS is proposed mainly for analytical purposes and other online algorithms may outperform FOSS in various perspectives. The following is a regret bound for FOSS, relying on the solution to Bellman equations.

**Definition 3** (Solution to the Bellman equations [57]). *Consider optimal control problem: $\min \lim_{N \to +\infty} \frac{1}{N} \sum_{t=0}^{N-1}(f(x_t) + g(u_t))$ where $x_{t+1} = Ax_t + Bu_t$. Let $\lambda^e$ be the optimal steady state cost $f(x^e) + g(u^e)$, which is also the optimal infinite-horizon average cost [56]. The Bellman equations for the problem is $h^e(x) + \lambda^e = \min_u(f(x) + g(u) + h^e(Ax + Bu))$. The solution to the Bellman equations, denoted by $h^e(x)$, is sometimes called as a bias function [57]. To ensure the uniqueness of the solution, some extra conditions, e.g. $h^e(0) = 0$, are usually imposed.*

**Theorem 2** (Regret bound of FOSS). *Let $(x_t^e, u_t^e)$ and $h_t^e(x)$ denote the optimal steady state and the bias function with respect to cost $f_t(x) + g_t(u)$ respectively for $0 \leq t \leq N - 1$. Suppose $h_t^e(x)$ exists for $0 \leq t \leq N - 1$,[4] then the regret of FOSS can be bounded by*

$$\text{Regret}(FOSS) = O\left( \sum_{t=0}^N (\|x_{t-1}^e - x_t^e\| + h_{t-1}^e(x_t^*) - h_t^e(x_t^*)) \right),$$

*where $\{x_t^*\}_{t=0}^N$ denotes the optimal state trajectory for (1), $x_{-1}^e = x_0^* = x_0 = 0$, $h_{-1}^e(x) = 0$, $h_N^e(x) = f_N(x)$, $x_N^e = \theta_N$. Consequently, by Theorem 1, the regret bound of RHTM with initialization FOSS is $\text{Regret}(RHTM) = O\left( \zeta^2(\frac{\sqrt{\zeta}-1}{\sqrt{\zeta}})^{2K} \sum_{t=0}^N (\|x_{t-1}^e - x_t^e\| + h_{t-1}^e(x_t^*) - h_t^e(x_t^*)) \right).$*

Theorem 2 bounds the regret by the variation of the optimal steady states $x_t^e$ and the bias functions $h_t^e$. If $f_t$ and $g_t$ do not change, $x_t^e$ and $h_t^e$ do not change, yielding a small $O(1)$ regret, i.e. $O(\|x_0^e\| + h_0^e(x_0))$, matching our intuition. Though Theorem 2 requires $h_t^e$ exists, the existence is guaranteed for many control problems, e.g. LQ tracking and control problems with turnpike properties [58, 22].

## 5 Linear quadratic tracking: regret upper bounds and a fundamental limit

To provide more intuitive meaning for our regret analysis in Theorem 1 and Theorem 2, we apply RHTM to the LQ tracking problem in Example 1. Results for the time varying $Q_t, R_t, \theta_t$ are provided in Appendix E; whereas here we focus on a special case which gives clean expressions for regret bounds: both an upper bound for RHTM with initialization FOSS and a lower bound for any online algorithm. Further, we show that the lower bound and the upper bound almost match each other, implying that our online algorithm RHTM uses the predictions in a nearly optimal way even though it only conducts a few gradient updates at each time step .

The special case of LQ tracking problems is in the following form,

$$\frac{1}{2} \sum_{t=0}^{N-1} \left[ (x_t - \theta_t)^\top Q (x_t - \theta_t) + u_t^\top R u_t \right] + \frac{1}{2} x_N^\top P^e x_N, \tag{9}$$

where $Q > 0$, $R > 0$, and $P^e$ is the solution to the algebraic Riccati equation with respect to $Q, R$ [59]. Basically, in this special case, $Q_t = Q$, $R_t = R$ for $0 \le t \le N - 1$, $Q_N = P^e$, $\theta_N = 0$, and only $\theta_t$ changes for $t = 0, 1, \ldots, N - 1$. The LQ tracking problem (9) aims to follow a time-varying trajectory $\{\theta_t\}$ with constant weights on the tracking cost and the control cost.

**Regret upper bound.** Firstly, based on Theorem 1 and Theorem 2, we have the following bound.

**Corollary 1.** *Under the stepsizes in Theorem 1, RHTM with FOSS as the initialization rule satisfies*

$$\text{Regret}(RHTM) = O\left( \zeta^2 (\frac{\sqrt{\zeta} - 1}{\sqrt{\zeta}})^{2K} \sum_{t=0}^{N} \|\theta_t - \theta_{t-1}\| \right)$$

*where $K = \lfloor (W - 1)/p \rfloor$, $\zeta$ is the condition number of the corresponding $C(\mathbf{z})$, $\theta_{-1} = 0$.*

This corollary shows that the regret can be bounded by the total variation of $\theta_t$ for constant $Q, R$.

**Fundamental limit.** For any online algorithm, we have the following lower bound.

**Theorem 3** (Lower Bound). *Consider $1 \le W \le N/3$, any condition number $\zeta > 1$, any variation budget $4\bar{\theta} \le L_N \le (2N + 1)\bar{\theta}$, and any controllability index $p \ge 1$. For any online algorithm $\mathcal{A}$, there exists an LQ tracking problem in form (9) where i) the canonical-form system $(A, B)$ has controllability index $p$, ii) the sequence $\{\theta_t\}$ satisfies the variation budget $\sum_{t=0}^{N} \|\theta_t - \theta_{t-1}\| \le L_N$, and iii) the corresponding $C(\mathbf{z})$ has condition number $\zeta$, such that the following lower bound holds*

$$\text{Regret}(\mathcal{A}) = \Omega\left( (\frac{\sqrt{\zeta} - 1}{\sqrt{\zeta} + 1})^{2K} L_N \right) = \Omega\left( (\frac{\sqrt{\zeta} - 1}{\sqrt{\zeta} + 1})^{2K} \sum_{t=0}^{N} \|\theta_t - \theta_{t-1}\| \right) \tag{10}$$

*where $K = \lfloor (W - 1)/p \rfloor$ and $\theta_{-1} = 0$.*

Firstly, the lower bound in Theorem 3 almost matches the upper bound in Corollary 1, especially when $\zeta$ is large, demonstrating that RHTM utilizes the predictions in a near-optimal way. The major conditions in Theorem 3 require that the prediction window is short compared with the horizon: $W \le N/3$, and the variation of the cost functions should not be too small: $L_N \ge 4\bar{\theta}$, otherwise the online control problem is too easy and the regret can be very small. Moreover, the small gap between the regret bounds is conjectured to be nontrivial, because this gap coincides with the long lasting gap in the convergence rate of the first-order algorithms for strongly convex and smooth optimization. In particular, the lower bound in Theorem 3 matches the fundamental convergence limit in [38], and the upper bound is by triple momentum's convergence rate, which is the best one to our knowledge.

## 6 Numerical experiments

**LQ tracking problem in Example 1.** The system considered here has $n = 2$, $m = 1$, and $p = 2$. More details of the experiment settings are provided in Appendix H. We compare RHGC with a suboptimal MPC algorithm, fast gradient MPC (subMPC) [23]. Roughly speaking, subMPC solves the $W$-stage truncated optimal control from $t$ to $t + W - 1$ by Nesterov's accelerated gradient [38], and one iteration of Nesterov's accelerated gradient requires $2W$ gradient evaluations of stage

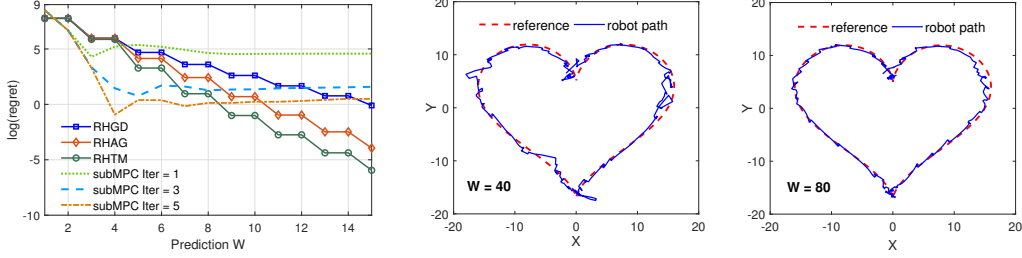

Figure 1: Regret for LQ tracking.      Figure 2: Two-wheel robot tracking with nonlinear dynamics.

cost function since $W$ stages are considered and each stage has two costs $f_t$ and $g_t$. This implies that, in terms of the number of gradient evaluations, subMPC with one iteration corresponds to our RHTM because RHTM also requires roughly $2W$ gradient evaluations per stage. Therefore, Figure 1 compares our RHGC algorithms with subMPC with one iteration. Figure 1 also plots subMPC with 3 and 5 iterations for more insights. Besides, Figure 1 plots not only RHGD and RHTM, but also RHAG, which is based on Nesterov's accelerated gradient. Figure 1 shows that all our algorithms achieve exponential decaying regrets with respect to $W$, and the regrets are piecewise constant, matching Theorem 1. Further, it is observed that RHTM and RHAG perform better than RHGD, which is intuitive because triple momentum and Nesterov's accelerated gradient are accelerated versions of gradient descent. In addition, our algorithms are much better than subMPC with 1 iteration, implying that our algorithms utilize the lookahead information more efficiently given similar computational time. Finally, subMPC achieves better performance by increasing the iteration number but the improvement saturates as $W$ increases, in contrast to the steady improvement of RHGC.

**Path tracking for a two-wheel mobile robot.** Though we presented our online algorithms on an LTI system, our RHGC methods are applicable to some nonlinear systems as well. Here we consider a two-wheel mobile robot with nonlinear kinematic dynamics $\dot{x} = v\cos\delta, \dot{y} = v\sin\delta, \dot{\delta} = w$ where $(x,y)$ is the robot location, $v$ and $w$ are the tangential and angular velocities respectively, $\delta$ denotes the tangent angle between $v$ and the $x$ axis [60]. The control is directly on the $v$ and $w$, e.g., via the pulse-width modulation (PWM) of the motor [61]. Given a reference path $(x_t^r, y_t^r)$, the objective is to balance the tracking performance and the control cost, i.e., $\min \sum_{t=0}^{N} \left[ c_t \cdot \left( (x_t - x_t^r)^2 + (y_t - y_t^r)^2 \right) + c_t^v \cdot (v_t)^2 + c_t^w \cdot (w_t)^2 \right]$. We discretize the dynamics with time interval $\Delta t = 0.025$s; then follow similar ideas in this paper to reformulate the optimal path tracking problem to an unconstrained optimization with respect to $(x_t, y_t)$ and apply RHGC. See Appendix H for details. Figure 2 plots the tracking results with window $W = 40$ and $W = 80$ corresponding to lookahead time 1s and 2s. A video showing the dynamic processes with different $W$ is provided at `https://youtu.be/fal56LTBD1s`. It is observed that the robot follows the reference trajectory well especially when the path is smooth but deviates a little more when the path has sharp turns, and a longer lookahead window leads to better tracking performance. These results confirm that our RHGC works effectively on nonlinear systems.

## 7 Conclusion

This paper studies the role of predictions on dynamic regrets of online control problems with linear dynamics. We design RHGC algorithms and provide regret upper bounds of two specific algorithms: RHGD and RHTM. We also provide a fundamental limit and show the fundamental limit almost matches RHTM's upper bound. This paper leads to many interesting future directions, some of which are briefly discussed below. The first direction is to study more realistic prediction models which considers random prediction noises, e.g. [33, 35, 62]. The second direction is to consider unknown systems with process noises, possibly by applying learning-based control tools [44, 46, 48]. Further, more studies could be conducted on general control problems including nonlinear control and control with input and state constraints. Besides, it is interesting to consider other performance metrics, such as competitive ratio, since the dynamic regret is non-vanishing. Finally, other future directions include closing the gap of the regret bounds and more discussion on the effect of the canonical-form transformation on the condition number.

## Acknowledgement

This work was supported by NSF Career 1553407, ARPA-E NODES, AFOSR YIP and ONR YIP programs.

## Footnotes

[1]The results in this paper can be extended to cost $c_t(x_t, u_t)$ with proper assumptions.

[4]$h_t^e$ may not be unique, so extra conditions can be imposed on $h_t^e$ for more interesting regret bounds.

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
