[Supplementary Material]

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

[5]This additional condition is for technical simplicity and can be removed.

[6] It is easy to generalize the construction to multi-input case by constructing $m$ decoupled subsystems.

[7]The proof can be easily generalized to random algorithms

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

## Appendices

In Appendix A, we will discuss the canonical-form transformation. In Appendix B, we will briefly introduce the triple momentum algorithm proposed in [39] and provide the proof of Theorem 1. In Appendix C, we will provide the proof of Lemma 1. In Appendix D, we will present the proof of Theorem 2. Appendix E provides a proof of Corollary 1 and regret analysis for more general linear quadratic tracking problems. Appendix F provides a proof of Theorem 3. In Appendix G, we will provide the proofs of the technical lemmas used in Appendix E. In Appendix H, we will provide a more detailed description of our numerical experiments.

## A    Canonical form

In this section, we introduce the linear transformation from a general LTI system to a canonical-form LTI system, and then discuss how to convert a general online optimal control problem to an online optimal control problem with a canonical-form system.

Firstly, consider a general LTI system: $x_{t+1} = Ax_t + Bu_t$ and two invertible matrices $S_x \in \mathbb{R}^n, S_u \in R^m$. Under the linear transformation on state and control: $\hat{x}_t = S_x x_t$, $\hat{u}_t = S_u u_t$, the equivalent LTI system with respect to the new state $\hat{x}_t$ and the new control $\hat{u}_t$ is

$$\hat{x}_{t+1} = S_x A S_x^{-1} \hat{x}_t + S_x B S_u^{-1} \hat{u}_t$$

By [40, Theorem 1], for any controllable $(A, B)$, there exist $S_x, S_u$ such that $\hat{A} = S_x A S_x^{-1}$ and $\hat{B} = S_x B S_u^{-1}$ are in the canonical form defined in Definition 1. The computation method of $S_x, S_u$ is also provided in [40].

In an online optimal control problem, since $A, B$ are known as priors, $S_x, \ S_u$ can be computed offline. When stage cost functions $f_t(x_t), g_t(u_t)$ are received online, the new cost functions $\hat{f}_t(\hat{x}_t), \hat{g}_t(\hat{u}_t)$ for the canonical-form system can be computed online by applying $S_x, S_u$:

$$\hat{f}_t(\hat{x}_t) = f_t(x_t) = f_t(S_x^{-1}\hat{x}_t), \quad \hat{g}_t(\hat{u}_t) = g_t(u_t) = g_t(S_u^{-1}\hat{u}_t)$$

Moreover, it is straightforward to verify that $\hat{f}_t(\hat{x}_t)$ and $\hat{g}_t(\hat{u}_t)$ still satisfy Assumption 2 and 3, just with perhaps different parameters. For example, $\hat{f}_t(\hat{x}_t)$ is $\mu_f / \|S_x\|^2$ strongly convex and $l_f \|S_x^{-1}\|^2$ smooth and $\hat{g}_t(\hat{u}_t)$ is convex and $l_g \|S_u^{-1}\|^2$ smooth. Therefore, it is without loss of generality to only consider online optimal control with canonical-form systems.

## B    Triple momentum and a proof of Theorem 1

Triple Momentum (TM) is an accelerated version of gradient descent proposed in [39]. When optimizing an unconstrained optimization $\min_{\mathbf{z}} C(\mathbf{z})$, at each iteration $j \geq 0$, TM conducts

$$\boldsymbol{\omega}(j+1) = (1+\delta_\omega)\boldsymbol{\omega}(j) - \delta_\omega\boldsymbol{\omega}(j-1) - \delta_c \nabla C(\mathbf{y}(j))$$
$$\mathbf{y}(j+1) = (1+\delta_y)\boldsymbol{\omega}(j+1) - \delta_y\boldsymbol{\omega}(j)$$
$$\mathbf{z}(j+1) = (1+\delta_z)\boldsymbol{\omega}(j+1) - \delta_z\boldsymbol{\omega}(j)$$

where $\boldsymbol{\omega}(j), \mathbf{y}(j)$ are auxiliary variables to accelerate the convergence, $\mathbf{z}(j)$ is the decision variable, $\boldsymbol{\omega}(0) = \boldsymbol{\omega}(-1) = \mathbf{z}(0) = \mathbf{y}(0)$ are given initial values.

Suppose $\mathbf{z} = (z_1^\top, \ldots, z_N^\top)^\top$. Zooming in to each coordinate $z_t$, the update of $z_t(j)$ by TM is provided below

$$\omega_t(j+1) = (1+\delta_\omega)\omega_t(j) - \delta_\omega\omega_t(j-1) - \delta_c \frac{\partial C}{\partial y_t}(\mathbf{y}(j))$$

$$y_t(j+1) = (1+\delta_y)\omega_t(j+1) - \delta_y\omega_t(j)$$
$$z_t(j+1) = (1+\delta_z)\omega_t(j+1) - \delta_z\omega_t(j)$$

By Section 3, $\frac{\partial C}{\partial y_t}(\mathbf{y}(j))$ only depends on stage cost functions and stage variables across a finite neighboring stages, allowing the online implementation in Algorithm 2 based on the finite-lookahead window.

TM enjoys a faster convergence rate than the gradient descent for $\mu_c$ strongly convex and $l_c$ smooth functions under proper step sizes. In particular, when $\gamma_c = \frac{1+\phi}{l_c}$, $\gamma_w = \frac{\phi^2}{2-\phi}$, $\gamma_y = \frac{\phi^2}{(1+\phi)(2-\phi)}$, $\gamma_z = \frac{\phi^2}{1-\phi^2}$, and $\phi = 1 - 1/\sqrt{\zeta}$, $\zeta = l_c/\mu_c$, by [39, Theorem 1], the convergence of TM satisfies:

$$C(\mathbf{z}(j)) - C(\mathbf{z}^*) \leq (\frac{\sqrt{\zeta}-1}{\sqrt{\zeta}})^{2j}\frac{l_c\zeta}{2}\|\mathbf{z}(0) - \mathbf{z}^*\|^2 \leq \zeta^2(\frac{\sqrt{\zeta}-1}{\sqrt{\zeta}})^{2j}(C(\mathbf{z}(0)) - C(\mathbf{z}^*)) \quad (11)$$

In the following, we will apply this result to prove Theorem 1.

## B.1 Proof of Theorem 1

By comparing TM with RHTM, it can be verified that $z_{t+1}(K)$ computed by RHTM is the same as $z_{t+1}(K)$ computed by the triple momentum after $K$ iterations. Moreover, according to the equivalence between the optimization $\min_{\mathbf{z}} C(\mathbf{z})$ and the optimal control $J(\mathbf{x}, \mathbf{u})$ in Lemma 1,

$$J(RHTM) = C(\mathbf{z}(K)), \ J(\varphi) = C(\mathbf{z}(0)), \ J^* = C(\mathbf{z}^*)$$

Finally, by utiltizing (11), the bound on $\text{Regret}(RHTM)$ is straightforward.

The regret of RHGD can be proved in the same way.

## C  Proof of Lemma 1

Property ii) and iii) can be directly verified by definition. Thus, it suffices to prove i): the strong convexity and smoothness of $C(\mathbf{z})$.

Notice that $x_t$, $u_t$ are linear with respect to $\mathbf{z}$ by (5) (6). For ease of reference, we define matrix $M^{x_t}, M^{u_t}$ to represent the relation between $x_t, u_t$ and $\mathbf{z}$, i.e, $x_t = M^{x_t}\mathbf{z}$ and $u_t = M^{u_t}\mathbf{z}$. Similarly, we write $\tilde{f}_t(z_{t-p+1}, \ldots, z_t)$ and $\tilde{g}_t(z_{t-p+1}, \ldots, z_{t+1})$ in terms of $\mathbf{z}$ for simplicity of notation:

$$\tilde{f}_t(z_{t-p+1}, \ldots, z_t) = \tilde{f}_t(\mathbf{z}) = f_t(M^{x_t}\mathbf{z})$$
$$\tilde{g}_t(z_{t-p+1}, \ldots, z_{t+1}) = \tilde{g}_t(\mathbf{z}) = g_t(M^{u_t}\mathbf{z})$$

A direct consequence of the linear relations is that $\tilde{f}_t(\mathbf{z})$ and $\tilde{g}_t(\mathbf{z})$ are convex with respect to $\mathbf{z}$ because $f_t(x_t), g_t(u_t)$ are convex and the linear transformation preserves convexity.

In the following, we will focus on the proof of strong convexity and smoothness. For simplicity, in the following, we only consider cost function $f_t, g_t$ with minimum values zero: $f_t(\theta_t) = 0$, and $g_t(\xi_t) = 0$ for all $t$. This is without loss of generality because by strong convexity and smoothness, $f_t, g_t$ have minimum values, and by subtracting the minimum value, we can let $f_t, g_t$ have minimum value 0.

**Strong convexity.** Since $\tilde{g}_t$ is convex, we only need to prove that $\sum_t \tilde{f}_t(\mathbf{z})$ is strongly convex then the sum $C(\mathbf{z})$ is strongly convex because the sum of convex functions and a strongly convex function is strongly convex.

In particular, by the strong convexity of $f_t(x_t)$, we have the following result: for any $\mathbf{z}, \mathbf{z}' \in \mathbb{R}^{Nm}$ and $x_t = M^{x_t}\mathbf{z}, x_t' = M^{x_t}\mathbf{z}'$:

$$\tilde{f}_t(\mathbf{z}') - \tilde{f}_t(\mathbf{z}) - \langle \nabla \tilde{f}_t(\mathbf{z}), \mathbf{z}' - \mathbf{z} \rangle - \frac{\mu_f}{2}\|z_t' - z_t\|^2$$

$$= \tilde{f}_t(\mathbf{z}') - \tilde{f}_t(\mathbf{z}) - \langle (M^{x_t})^\top \nabla f_t(x_t), \mathbf{z}' - \mathbf{z} \rangle - \frac{\mu_f}{2}\|z_t' - z_t\|^2$$

$$= \tilde{f}_t(\mathbf{z}') - \tilde{f}_t(\mathbf{z}) - \langle \nabla f_t(x_t), M^{x_t}(\mathbf{z}' - \mathbf{z}) \rangle - \frac{\mu_f}{2}\|z_t' - z_t\|^2$$

$$= \tilde{f}_t(\mathbf{z}') - \tilde{f}_t(\mathbf{z}) - \langle \nabla f_t(x_t), x_t' - x_t \rangle - \frac{\mu_f}{2}\|z_t' - z_t\|^2$$

$$\geq f_t(x_t') - f_t(x_t) - \langle \nabla f_t(x_t), x_t' - x_t \rangle - \frac{\mu_f}{2}\|x_t' - x_t\|^2 \geq 0$$

where the first equality is by the chain rule, the second equality is by the definition of inner product, the third equality is by the definition of $x_t, x_t'$, the first inequality is by $\tilde{f}_t(z) = f_t(x)$ and $z_t = (x_t^{k_1}, \ldots, x_t^{k_m})^\top$, and the last inequality is because $f_t(x_t)$ is $\mu_f$ strongly convex.

Summing over $t$ on both sides of the inequality results in the strong convexity of $\sum_t \tilde{f}_t(\mathbf{z})$:

$$\sum_{t=1}^{N} \left[ \tilde{f}_t(\mathbf{z}') - \tilde{f}_t(\mathbf{z}) - \langle \nabla \tilde{f}_t(\mathbf{z}), \mathbf{z}' - \mathbf{z} \rangle - \frac{\mu_f}{2} \|z_t' - z_t\|^2 \right]$$

$$= \sum_{t=1}^{N} \tilde{f}_t(\mathbf{z}') - \sum_{t=1}^{N} \tilde{f}_t(\mathbf{z}) - \langle \nabla \sum_{t=1}^{N} \tilde{f}_t(\mathbf{z}), \mathbf{z}' - \mathbf{z} \rangle - \frac{\mu_f}{2} \|\mathbf{z}' - \mathbf{z}\|^2 \geq 0$$

Consequently, $C(\mathbf{z})$ is strongly convex with parameter at least $\mu_f$ by the convexity of $\tilde{g}_t$.

**Smoothness.** We will prove the smoothness by considering $\tilde{f}_t(\mathbf{z})$ and $\tilde{g}_t(\mathbf{z})$ respectively.

Firstly, let's consider $\tilde{f}_t(\mathbf{z})$. Similar to the proof for strong convexity, we use the smoothness of $f_t(x_t)$. For any $\mathbf{z}, \mathbf{z}'$, and $x_t = M^{x_t}\mathbf{z}$, $x_t' = M^{x_t}\mathbf{z}'$, we can show that

$$\tilde{f}_t(\mathbf{z}') = f_t(x_t') \leq f_t(x_t) + \langle \nabla f_t(x_t), x_t' - x_t \rangle + \frac{l_f}{2} \|x_t' - x_t\|^2$$

$$\leq \tilde{f}_t(\mathbf{z}) + \langle \nabla \tilde{f}_t(\mathbf{z}), \mathbf{z}' - \mathbf{z} \rangle + \frac{l_f}{2} (\|z_{t-p+1}' - z_{t-p+1}\|^2 + \cdots + \|z_t' - z_t\|^2)$$

where the second inequality is by $x_t = M^{x_t}\mathbf{z}$ and the chain rule and (5).

Secondly, we consider $\tilde{g}_t(z)$ in a similar way. For any $\mathbf{z}, \mathbf{z}'$, and $u_t = M^{u_t}\mathbf{z}$, $u_t' = M^{u_t}\mathbf{z}'$, we have

$$\tilde{g}_t(\mathbf{z}') = g_t(u_t') \leq g_t(u_t) + \langle \nabla g_t(u_t), u_t' - u_t \rangle + \frac{l_g}{2} \|u_t' - u_t\|^2$$

$$= \tilde{g}_t(\mathbf{z}) + \langle (M^{u_t})^\top \nabla g_t(u_t), \mathbf{z}' - \mathbf{z} \rangle + \frac{l_g}{2} \|u_t' - u_t\|^2$$

$$= \tilde{g}_t(\mathbf{z}) + \langle \nabla \tilde{g}_t(\mathbf{z}), \mathbf{z} - \mathbf{z} \rangle + \frac{l_g}{2} \|u_t' - u_t\|^2$$

Since $u_t = z_{t+1} - A(\mathcal{I}, :)x_t = [I_m, -A(\mathcal{I}, :)](z_{t+1}^\top, x_t^\top)^\top$, we have that

$$\frac{l_g}{2} \|u_t' - u_t\|^2 \leq \frac{l_g}{2} \|[I_m, -A(\mathcal{I}, :)] \left[ ((z_{t+1}')^\top, (x_t')^\top)^\top - (z_{t+1}^\top, x_t^\top)^\top \right] \|^2$$

$$\leq \frac{l_g}{2} \|[I_m, -A(\mathcal{I}, :)]\|^2 (\|z_{t+1} - z_{t+1}'\|^2 + \|x_t - x_t'\|^2)$$

$$\leq \frac{l_g}{2} \|[I_m, -A(\mathcal{I}, :)]\|^2 (\|z_{t+1} - z_{t+1}'\|^2 + \cdots + \|z_{t-p+1} - z_{t-p+1}'\|^2)$$

Finally, by summing $\tilde{f}_t(\mathbf{z}'), \tilde{g}_t(\mathbf{z}')$'s inequalities above over all $t$, we have

$$C(\mathbf{z}') \leq C(\mathbf{z}) + \langle \nabla C(\mathbf{z}), \mathbf{z}' - \mathbf{z} \rangle + (pl_f + (p+1)l_g\|[I_m, -A(\mathcal{I}, :)]\|^2)/2\|\mathbf{z}' - \mathbf{z}\|^2$$

Thus, we have proved the smoothness of $C(\mathbf{z})$.

## D   Proof of Theorem 2

Remember that $\text{Regret}(FOSS) = J(FOSS) - J^*$. To bound the regret, we let the sum of the optimal steady state costs, $\sum_{t=0}^{N-1} \lambda_t^e$, be a middle ground and bound $J(FOSS) - \sum_{t=0}^{N-1} \lambda_t^e$ and $\sum_{t=0}^{N-1} \lambda_t^e - J^*$ in Lemma 2 and Lemma 3 respectively. Then, the regret bound can be obtained by combining the two bounds.

**Lemma 2** (Bound on $J(FOSS) - \sum_{t=0}^{N-1} \lambda_t^e$). *Let $x_t(0)$ denote the state determined by FOSS.*

$$J(FOSS) - \sum_{t=0}^{N-1} \lambda_t^e \leq c_1 \sum_{t=0}^{N-1} \|x_t^e - x_{t-1}^e\| + f_N(x_N(0)) = O\left( \sum_{t=0}^{N} \|x_t^e - x_{t-1}^e\| \right)$$

*where we define $x_N^e := \delta_N$, $x_{-1}^e := x_0 = 0$ for simplicity of notation, $c_1$ is a constant that does not depend on $N, W$ and big $O$ hides a constant that does not depend on $N, W$.*

**Lemma 3** (Bound on $\sum_{t=0}^{N-1} \lambda_t^e - J^*$). *Let $h_t^e(x)$ denote a solution to the Bellman equations under cost $f_t(x) + g_t(u)$. Let $\{x_t^*\}$ denote the optimal state trajectory to the offline optimal control (1).*

$$\sum_{t=0}^{N-1} \lambda_t^e - J^* \leq \sum_{t=1}^{N}(h_{t-1}^e(x_t^*) - h_t^e(x_t^*)) - h_0^e(x_0) = \sum_{t=0}^{N}(h_{t-1}^e(x_t^*) - h_t^e(x_t^*))$$

*where we define $h_N^e(x) := f_N(x)$, $h_{-1}^e(x) := 0$ and $x_0^* := x_0$ for simplicity of notation.*

Then, we can complete the proof by applying Lemma 2 and 3:

$$J(FOSS) - J^* = J(FOSS) - \sum_{t=0}^{N-1} \lambda_t^e + \sum_{t=0}^{N-1} \lambda_t^e - J^*$$

$$= O\left(\sum_{t=0}^{N}(\|x_{t-1}^e - x_t^e\| + h_{t-1}^e(x_t^*) - h_t^e(x_t^*))\right)$$

In the following, we will prove Lemma 2 and 3 respectively. For simplicity, we only consider cost function $f_t, g_t$ with minimum values zero: $f_t(\theta_t) = 0$, and $g_t(\xi_t) = 0$ for all $t$. There is no loss of generality because by strong convexity and smoothness, $f_t, g_t$ have minimum values, and by subtracting the minimum value, we can let $f_t, g_t$ have minimum value 0.

### D.1  Proof of Lemma 2.

Notice that $J(FOSS) = \sum_{t=0}^{N-1}(f_t(x_t(0)) + g_t(u_t(0))) + f_N(x_N(0))$ and $\sum_{t=0}^{N-1} \lambda_t^e = \sum_{t=0}^{N-1}(f_t(x_t^e) + g_t(u_t^e))$. Thus, it suffices to bound $f_t(x_t(0)) - f_t(x_t^e)$ and $g_t(u_t(0)) - g_t(u_t^e)$ for $0 \leq t \leq N - 1$. We will first focus on $f_t(x_t(0)) - f_t(x_t^e)$, then bound $g_t(u_t(0)) - g_t(u_t^e)$ in the same way.

For $0 \leq t \leq N - 1$, by the convexity of $f_t$, and the property of $L_2$ norm,

$$f_t(x_t(0)) - f_t(x_t^e) \leq \langle \nabla f_t(x_t(0)), x_t(0) - x_t^e \rangle \leq \|\nabla f_t(x_t(0))\| \|x_t(0) - x_t^e\| \quad (12)$$

In the following, we will bound $\|\nabla f_t(x_t(0))\|$ and $\|x_t(0) - x_t^e\|$ respectively.

Firstly, we provide a bound on $\|\nabla f_t(x_t(0))\|$:

$$\|\nabla f_t(x_t(0))\| = \|\nabla f_t(x_t(0)) - \nabla f_t(\theta_t)\| \leq l_f \|x_t(0) - \theta_t\| \leq l_f(\sqrt{n}\bar{x}^e + \bar{\theta}) \quad (13)$$

where the first equality is because $\theta_t$ is the global minimizer of $f_t$, and first inequality is by Lipschitz smoothness, the second inequality is by $\|\theta_t\| \leq \bar{\theta}$ according to Assumption 3 and by $\|x_t(0)\| \leq \sqrt{n}\bar{x}^e\|$ proved in the following lemma.

**Lemma 4** (Uniform upper bounds on $x_t^e, u_t^e, x_t(0), u_t(0)$). *There exist $\bar{x}^e$ and $\bar{u}^e$ that are independent of $N, W$, such that $\|x_t^e\| \leq \bar{x}^e$ and $\|u_t^e\| \leq \bar{u}^e$ for all $0 \leq t \leq N - 1$. Moreover, $\|x_t(0)\| \leq \sqrt{n}\bar{x}^e$ for $0 \leq t \leq N$ and $\|u_t(0)\| \leq \sqrt{n}\bar{u}^e$ for $0 \leq t \leq N - 1$, where $x_t(0), u_t(0)$ denote the state and control at $t$ determined by FOSS.*

The proof is technical and is deferred to Appendix D.3.

Secondly, we provide a bound on $\|x_t(0) - x_t^e\|$. The proof relies on the expressions of the steady state $x_t^e$ and the initialized state $x_t(0)$ of a canonical-form system.

**Lemma 5** (The steady state and the initialized state of canonical-form systems). *Consider a canonical-form system: $x_{t+1} = Ax_t + Bu_t$.*

*(a) Any steady state $(x, u)$ is in the form of*

$$x = (\underbrace{z^1, \ldots, z^1}_{p_1}, \underbrace{z^2, \ldots, z^2}_{p_2}, \ldots, \underbrace{z^m, \ldots, z^m}_{p_m})^\top$$

$$u = (z^1, \ldots, z^m)^\top - A(\mathcal{I}, :)x$$

*for some $z^1, \ldots, z^m \in \mathbb{R}$. Let $z = (z^1, \ldots, z^m)^\top$. For the optimal steady state with respect to cost $f_t + g_t$, we denote the corresponding $z$ as $z_t^e$, and the optimal steady state can be represented as $x_t^e = (z_t^{e,1}, \ldots, z_t^{e,1}, z_t^{e,2}, \ldots, z_t^{e,2}, \ldots, z_t^{e,m}, \ldots, z_t^{e,m})^\top$ and $u_t^e = z_t^e - A(\mathcal{I}, :)x_t^e$ for $0 \leq t \leq N - 1$.*

*(b) By FOSS initialization, $z_{t+1}(0) = z_t^e$, and $x_t(0)$, $u_t(0)$ satisfy*

$$x_t(0) = (\underbrace{z_{t-p_1}^{e,1}, \ldots, z_{t-1}^{e,1}}_{p_1}, \underbrace{z_{t-p_2}^{e,2}, \ldots, z_{t-1}^{e,2}}_{p_2}, \ldots, \underbrace{z_{t-p_m}^{e,m}, \ldots, z_{t-1}^{e,m}}_{p_m}), \qquad 0 \le t \le N$$

$$u_t(0) = z_t^e - A(\mathcal{I}, :)x_t(0) \qquad\qquad\qquad\qquad\qquad 0 \le t \le N - 1$$

*where $z_t^e = 0$ for $t \le -1$.*

*Proof.* (a) This is by the definition of the canonical form and the definition of the steady state.

(b) By the initialization, $z_t(0) = x_{t-1}^{e,\mathcal{I}} = z_{t-1}^e$. By the relation between $z_t(0)$ and $x_t(0)$, $u_t(0)$, we have $x_t^{\mathcal{I}}(0) = z_t(0) = z_{t-1}^e$, and $x_t^{\mathcal{I}-1}(0) = z_{t-1}(0) = z_{t-2}^e$, so on and so forth. This proves the structure of $x_t(0)$. The structure of $u_t(0)$ is because $u_t(0) = z_{t+1}(0) - A(\mathcal{I}, :)x_t(0) = z_t^e - A(\mathcal{I}, :)x_t(0)$

$\square$

By Lemma 5, we can bound $\|x_t(0) - x_t^e\|$ for $0 \le t \le N - 1$ by

$$\|x_t(0) - x_t^e\| \le \sqrt{\|z_{t-1}^e - z_t^e\|^2 + \cdots + \|z_{t-p}^e - z_t^e\|^2}$$

$$\le \sqrt{\|x_{t-1}^e - x_t^e\|^2 + \cdots + \|x_{t-p}^e - x_t^e\|^2}$$

$$\le \|x_{t-1}^e - x_t^e\| + \cdots + \|x_{t-p}^e - x_t^e\|$$

$$\le p(\|x_{t-1}^e - x_t^e\| + \cdots + \|x_{t-p}^e - x_{t-p+1}^e\|) \tag{14}$$

Combining (12) (13) and (14) yields

$$\sum_{t=0}^{N-1} f_t(x_t(0)) - f_t(x_t^e) \le \sum_{t=0}^{N-1} \|\nabla f_t(x_t(0))\| \|x_t(0) - x_t^e\|$$

$$\le \sum_{t=0}^{N-1} l_f(\sqrt{n}\bar{x}^e + \bar\theta)p(\|x_{t-1}^e - x_t^e\| + \cdots + \|x_{t-p}^e - x_{t-p+1}^e\|)$$

$$\le p^2 l_f(\sqrt{n}\bar{x}^e + \bar\theta) \sum_{t=0}^{N-1} \|x_{t-1}^e - x_t^e\| \tag{15}$$

Notice that the constant term $p^2 l_f(\sqrt{n}\bar{x}^e + \bar\theta)$ does not depend on $N, W$.

Similarly, we can provide a bound on $g_t(u_t(0)) - g_t(u_t^e)$.

$$\sum_{t=0}^{N-1} g_t(u_t(0)) - g_t(u_t^e) \le \sum_{t=0}^{N-1} \|\nabla g_t(u_t(0))\| \|u_t(0) - u_t^e\|$$

$$\le \sum_{t=0}^{N-1} l_g \|u_t(0) - \xi_t\| \|u_t(0) - u_t^e\|$$

$$\le \sum_{t=0}^{N-1} l_g(\sqrt{n}\bar{u}^e + \bar\xi) \|A(\mathcal{I}, :)x_t(0) - A(\mathcal{I}, :)x_t^e\|$$

$$\le \sum_{t=0}^{N-1} l_g(\sqrt{n}\bar{u}^e + \bar\xi) \|A(\mathcal{I}, :)\| \|x_t(0) - x_t^e\|$$

$$\le p^2 l_g(\sqrt{n}\bar{u}^e + \bar\xi) \|A(\mathcal{I}, :)\| \sum_{t=0}^{N-1} \|x_{t-1}^e - x_t^e\| \tag{16}$$

where the first inequality is by the convexity, the second inequality is because $\xi_t$ is the global minimizer of $g_t$ and $g_t$ is $l_g$-smooth, the third inequality is by Assumption 3, Lemma 4 and Lemma

5, the fifth inequality is by (14). Notice that the constant term $p^2 l_g(\sqrt{n}\bar{u}^e + \bar{\xi})\|A(\mathcal{I},:)\|$ does not depend on $N, W$.

By (15) and (16), we complete the proof of the first inequality in the statement of Lemma 2:

$$J(FOSS) - \sum_{t=0}^{N-1} \lambda_t^e \leq c_1 \sum_{t=0}^{N-1} \|x_{t-1}^e - x_t^e\| + f_N(x_N(0))$$

where $c_1$ does not depend on $N, W$.

By defining $x_N^e = \theta_N$, we can bound $f_N(x_N(0))$ by $\|x_N(0) - x_N^e\|$ up to some constants because $f_N(x_N(0)) = f_N(x_N(0)) - f_N(\theta_N) \leq \frac{l_f}{2}(\sqrt{n}\bar{x}^e + \bar{\theta})\|x_N(0) - x_N^e\|$. By the same argument as in (14), we have $\|x_N(0) - x_N^e\| = O(\sum_{t=0}^{N} \|x_{t-1}^e - x_t^e\|)$, where the big $O$ hides some constant that does not depend on $N, W$. Consequently,

$$J(FOSS) - \sum_{t=0}^{N-1} \lambda_t^e = O\left(\sum_{t=0}^{N} \|x_{t-1}^e - x_t^e\|\right)$$

$\square$

## D.2   Proof of Lemma 3.

The proof heavily relies on dynamic programming and the Bellman equations. For simplicity, we introduce a Bellman operator $\mathcal{B}(f+g, h)$ defined by $\mathcal{B}(f+g, h)(x) = \min_u(f(x) + g(u) + h(Ax + Bu))$. Now the Bellman equations can be written as $\mathcal{B}(f+g, h^e)(x) = h^e(x) + \lambda^e$ for any $x$.

We define a sequence of auxiliary functions $S_k$: $S_k(x) = h_k^e(x) + \sum_{t=k}^{N-1} \lambda_t^e$ for $k = 0, \dots, N$, where $h_N^e(x) = f_N(x)$.

We first provide a recursive equation for $S_k$. By Bellman equations, we have $h_k^e(x) + \lambda_k^e = \mathcal{B}(f_k + g_k, h_k^e)(x)$ for $0 \leq k \leq N - 1$. Let $\pi_k^e$ be the corresponding optimal control policy that solves the Bellman equations. We have the following recursive relation for $S_k$ when $0 \leq k \leq N - 1$:

$$S_k(x) = \mathcal{B}(f_k + g_k, S_{k+1} - h_{k+1}^e + h_k^e)(x)$$

where $S_N(x) = f_N(x)$.

Further, let $V_k(x)$ denote the optimal cost-to-go function from $k$ to $N$, then we obtain a recursive equation for $V_k$ by dynamic programming:

$$V_k(x) = \mathcal{B}(f_k + g_k, V_{k+1})(x) = f_k(x) + g_k(\pi_k^*(x)) + V_{k+1}(Ax + B\pi_k^*(x))$$

where $0 \leq k \leq N - 1$, and $\pi_k^*$ denotes the optimal control policy and $V_N(x) = f_N(x)$.

Now, we are ready for a recursive inequality for $S_k(x_k^*) - V_k(x_k^*)$. Let $\{x_k^*\}$ denote the optimal trajectory, then $x_{k+1}^* = Ax_k^* + B\pi_k^*(x_k^*)$. For any $k = 0, \dots, N - 1$,

$$\begin{aligned}
S_k(x_k^*) - V_k(x_k^*) &= \mathcal{B}(f_k + g_k, S_{k+1} - h_{k+1}^e + h_k^e)(x_k^*) - \mathcal{B}(f_k + g_k, V_{k+1})(x_k^*) \\
&\leq f_k(x_k^*) + g_k(\pi_k^*(x_k^*)) + S_{k+1}(x_{k+1}^*) - h_{k+1}^e(x_{k+1}^*) + h_k^e(x_{k+1}^*) \\
&\quad - (f_k(x_k^*) + g_k(\pi_k^*(x_k^*)) + V_{k+1}(x_{k+1}^*)) \\
&= S_{k+1}(x_{k+1}^*) - h_{k+1}^e(x_{k+1}^*) + h_k^e(x_{k+1}^*) - V_{k+1}(x_{k+1}^*)
\end{aligned}$$

where the first inequality is because $\pi_k^*$ is not optimal for the Bellman operator $\mathcal{B}(f_k + g_k, S_{k+1} - h_{k+1}^e + h_k^e)(x_k^*)$.

Summing over $k = 0, \dots, N - 1$ the recursive inequality for $S_k(x_k^*) - V_k(x_k^*)$ yields

$$S_0(x_0) - V_0(x_0) \leq \sum_{k=0}^{N-1} (h_k^e(x_{k+1}^*) - h_{k+1}^e(x_{k+1}^*))$$

By subtracting $h_0^e(x_0)$ on both sides,

$$\sum_{t=0}^{N-1} \lambda_t^e - J^* \leq \sum_{k=0}^{N-1} (h_k^e(x_{k+1}^*) - h_{k+1}^e(x_{k+1}^*)) - h_0^e(x_0)$$

For the simplicity of notation, we define $h^e_{-1}(x_0) = 0$ and $x^*_0 = x_0$, then the bound can be written as

$$\sum_{t=0}^{N-1} \lambda^e_t - J^* \leq \sum_{k=0}^{N} (h^e_{k-1}(x^*_k) - h^e_k(x^*_k))$$

□

### D.3 Proof of Lemma 4

The proof relies on the (strong) convexity and smoothness of the cost functions and the uniform upper bounds on $\theta_t, \xi_t$.

First of all, suppose there exists $\bar{x}^e$ such that $\|x^e_t\|_2 \leq \bar{x}^e$ for all $0 \leq t \leq N - 1$. We will bound $u^e_t, x_t(0), u_t(0)$ by using $\bar{x}^e$. Notice that the optimal steady state and the corresponding steady control satisfy: $u^e_t = x^{e,\mathcal{I}}_t - A(\mathcal{I},:)x^e_t$. If we can bound $x^e_t$ by $\|x^e_t\| \leq \bar{x}^e$ for all $t$, $u^e_t$ can be bounded accordingly:

$$\|u^e_t\| \leq \|x^{e,\mathcal{I}}_t\| + \|A(\mathcal{I},:)x^e_t\| \leq \|x^e_t\| + \|A(\mathcal{I},:)\|\|x^e_t\| \leq (1 + \|A(\mathcal{I},:)\|)\bar{x}^e =: \bar{u}^e$$

Moreover, $x_t(0)$ can also be bounded by $\bar{x}^e$ multiplied by some factors, because by Lemma 5, $x_t(0)$'s each entry is determined by some entry of $x^e_s$ for $s < t$. As a result, for $0 \leq t \leq N$

$$\|x_t(0)\|_2 \leq \sqrt{n}\|x_t(0)\|_\infty \leq \sqrt{n} \max_{s<t} \|x^e_s\|_\infty \leq \sqrt{n} \max_{s<t} \|x^e_s\|_2 \leq \sqrt{n}\bar{x}^e$$

We can bound $u_t(0)$ by noticing that $u_t(0) = x^{\mathcal{I}}_{t+1}(0) - A(\mathcal{I},:)x_t(0)$ and

$$\|u_t(0)\| \leq \|x^{\mathcal{I}}_{t+1}(0)\| + \|A(\mathcal{I},:)x_t(0)\| \leq \|x_{t+1}(0)\| + \|A(\mathcal{I},:)\|\|x_t(0)\|$$
$$\leq (1 + \|A(\mathcal{I},:)\|)\sqrt{n}\bar{x}^e = \sqrt{n}\bar{u}^e$$

Next, it suffices to prove $\|x^e_t\| \leq \bar{x}^e$ for all $t$ for some $\bar{x}^e$. To prove this bound, we construct another (suboptimal) steady state: $\hat{x}_t = (\theta^1_t, \ldots, \theta^1_t)$. Let $\hat{u}_t = \hat{x}^{\mathcal{I}}_t - A(\mathcal{I},:)\hat{x}_t$. It can be easily verified that $(\hat{x}_t, \hat{u}_t)$ is indeed a steady state of the canonical-form system. Moreover, $\hat{x}_t$ and $\hat{u}_t$ can be bounded similarly as follows.

$$\|\hat{x}_t\| \leq \sqrt{n}|\theta^1_t| \leq \sqrt{n}\|\theta_t\|_\infty \leq \sqrt{n}\|\theta_t\| \leq \sqrt{n}\bar{\theta}$$
$$\|\hat{u}_t\|_2 \leq (1 + \|A(\mathcal{I},:)\|)\|\hat{x}_t\| \leq (1 + \|A(\mathcal{I},:)\|)\sqrt{n}\bar{\theta}$$

Now, we can bound $\|x^e_t - \theta_t\|$.

$$\frac{\mu}{2}\|x^e_t - \theta_t\|^2 \leq f_t(x^e_t) - f_t(\theta_t) + g_t(u^e_t) - g_t(\xi_t)$$
$$\leq f_t(\hat{x}_t) - f_t(\theta_t) + g_t(\hat{u}_t) - g_t(\xi_t)$$
$$\leq \frac{l_f}{2}\|\hat{x}_t - \theta_t\|^2 + \frac{l_g}{2}\|\hat{u}_t - \xi_t\|^2$$
$$\leq l_f(\|\hat{x}_t\|^2 + \|\theta_t\|^2) + l_g(\|\hat{u}_t\|^2 + \|\xi_t\|^2)$$
$$\leq l_f(n\bar{\theta}^2 + \bar{\theta}^2) + l_g(((1 + \|A(\mathcal{I},:)\|)\sqrt{n}\bar{\theta})^2 + \bar{\xi}) =: c_5$$

where the first inequality is by $f_t$'s strong convexity and $g_t$'s convexity, the second inequality is because $(x^e_t, u^e_t)$ is an optimal steady state, the third inequality is by the smoothness and $\nabla f_t(\theta_t) = \nabla g_t(\xi_t) = 0$, the last inequality is by the bounds of $\|\hat{x}_t\|, \|\hat{u}_t\|, \theta_t$, and $\xi_t$.

As a result, we have $\|x^e_t - \theta_t\| \leq \sqrt{2c_5/\mu}$. Then, we can bound $x^e_t$ by $\|x^e_t\| \leq \|\theta_t\| + \sqrt{2c_5/\mu} \leq \bar{\theta} + \sqrt{2c_5/\mu} =: \bar{x}^e$ for all $t$. It can be verified that $\bar{x}^e$ does not depend on $N, W$.

□

## E  Linear quadratic tracking

In this section, we will provide a regret bound in Corollary 2 for the general LQT defined in Example 1. Based on this, we prove Corollary 1, which is a special case when $Q_t, R_t$ are not changing.

## E.1 Regret bound on the general online LQT problems

Before the regret bound, we provide an important lemma to characterize the solution to the Bellman equations of the LQT problem.

**Lemma 6.** *One solution to the Bellman equations with stage cost $\frac{1}{2}(x-\theta)^\top Q(x-\theta) + \frac{1}{2}u^\top Ru$ can be represented by*

$$h^e(x) = \frac{1}{2}(x - \beta^e)^\top P^e(x - \beta^e) \tag{17}$$

*where $P^e$ denotes the solution to the discrete-time algebraic Riccati equation (DARE) with respect to $Q, R, A, B$*

$$P^e = Q + A^\top(P^e - P^e B(B^\top P^e B + R)^{-1} B^\top P^e)A \tag{18}$$

*and $\beta^e = F\theta$ where $F$ is a matrix determined by $A, B, Q, R$.*

The proof is in Appendix G.

For simplicity of notation, let $P^e(Q, R)$ denote the solution to the DARE under the parameters $Q, R, A, B$ and $F(Q, R)$ denote the matrix in $\beta^e = F\theta$ given parameters $Q, R, A, B$. Here we omit $A, B$ in the arguments of the functions because they will not change in this paper.

In addition, we introduce the following useful notations: $\underline{Q} = \mu_f I_n, \bar{Q} = l_f I_n, \underline{R} = \mu_g I_m, \bar{R} = l_g I_m$ for $\mu_f, \mu_g > 0, 0 < l_f, l_g < +\infty$; and $\bar{P} = P^e(\bar{Q}, \bar{R})$ and $\underline{P} = P^e(\underline{Q}, \underline{R})$. Based on the notations above, we define some sets of matrices to be used later:

$$\mathcal{Q} = \{Q \mid \underline{Q} \le Q \le \bar{Q}\},$$
$$\mathcal{R} = \{R \mid \underline{R} \le R \le \bar{R}\},$$
$$\mathcal{P} = \{P \mid \underline{P} \le P \le \bar{P}\}.$$

Now, we are ready for the regret bound for the general LQT problem.

**Corollary 2** (Bound on general LQT). *Consider the LQT problem in Example 1. Suppose for $t = 0, 1, \ldots, N-1$, the cost matrices satisfy $Q_t \in \mathcal{Q}$, $R_t \in \mathcal{R}$. Suppose the terminal cost function satisfies $Q_N \in \mathcal{P}$.[5] Then, the regret of RHTM with initialization FOSS can be bounded by*

$$\mathrm{Regret}(RHTM) = O\left(\zeta^2(\frac{\sqrt{\zeta}-1}{\sqrt{\zeta}})^{2K}\left(\sum_{t=1}^N(\|P_t^e - P_{t-1}^e\| + \|\beta_t^e - \beta_{t-1}^e\|) + \sum_{t=0}^N \|x_{t-1}^e - x_t^e\|\right)\right)$$

*where $K = \lfloor(W-1)/p\rfloor$, $x_{-1}^e = x_0$, $x_N^e = \theta_N$, $\zeta$ is the condition number of the corresponding $C(\mathbf{z})$, $(x_t^e, u_t^e)$ is the optimal steady state under cost $Q_t, R_t, \theta_t$, $P_t^e = P^e(Q_t, R_t)$ and $\beta_t^e = F(Q_t, R_t)\theta_t$ for $t = 0, \ldots, N-1$ and $\beta_N^e = \theta_N$, $P_N^e = Q_N$.*

*Proof.* Before the proof, we introduce some supportive lemmas on the uniform bounds of $P_t^e, \beta_t^e, x_t^*$ respectively. The intuition behind these uniform bounds is that the cost function coefficients $Q_t, R_t, \theta_t$ are all uniformly bounded by Assumption 2 and 3. The proofs are technical and deferred to Appendix G.

**Lemma 7** (Upper bound on $x_t^*$). *For any $Q_t \in \mathcal{Q}, R_t \in \mathcal{R}, Q_N \in \mathcal{P}$, there exists $\bar{x}$ that does not depend on $t, N, W$, such that*

$$\|x_t^*\|_2 \le \bar{x}, \qquad \forall\, 0 \le t \le N.$$

**Lemma 8** (Upper bound on $\beta^e$). *For any $Q \in \mathcal{Q}, R \in \mathcal{R}$, any $\|\theta\| \le \bar{\theta}$, there exists $\bar{\beta} \ge 0$ that does not depend on $N$ and only depends on $A, B, l_f, \mu_f, l_g, \mu_g, \bar{\theta}$, such that $\max(\bar{\theta}, \|\beta^e\|) \le \bar{\beta}$, where $\beta^e$ is defined in Lemma 6.*

**Lemma 9** (Upper bound on $P^e$). *For any $Q \in \mathcal{Q}, R \in \mathcal{R}$, we have $P^e = P^e(Q, R) \in \mathcal{P}$. Consequently, $\|P^e\| \le \upsilon_{\max}(\bar{P})$, where $\upsilon_{\max}(\bar{P})$ denotes the largest eigenvalue of $\bar{P}$.*

Now, we are ready for the proof of Corollary 2.

By Theorem 2, we only need to bound $\sum_{t=0}^{N}(h_{t-1}^e(x_t^*) - h_t^e(x_t^*))$. By definition, $P_N^e = Q_N, \beta_N^e = \theta_N, h_N^e(x) = f_N(x)$, so we can write $h_t^e(x) = \frac{1}{2}(x - \beta_t^e)^\top P_t^e(x - \beta_t^e)$ for $0 \le t \le N$.

For $0 \le t \le N - 1$, we split $h_t^e(x_{t+1}^*) - h_{t+1}^e(x_{t+1}^*)$ into two parts.

$$
\begin{aligned}
h_t^e(x_{t+1}^*) - h_{t+1}^e(x_{t+1}^*) &= \frac{1}{2}(x_{t+1}^* - \beta_t^e)^\top P_t^e(x_{t+1}^* - \beta_t^e) - \frac{1}{2}(x_{t+1}^* - \beta_{t+1}^e)^\top P_{t+1}^e(x_{t+1}^* - \beta_{t+1}^e) \\
&= \underbrace{\frac{1}{2}(x_{t+1}^* - \beta_t^e)^\top P_t^e(x_{t+1}^* - \beta_t^e) - \frac{1}{2}(x_{t+1}^* - \beta_{t+1}^e)^\top P_t^e(x_{t+1}^* - \beta_{t+1}^e)}_{\text{Part 1}} \\
&\quad + \underbrace{\frac{1}{2}(x_{t+1}^* - \beta_{t+1}^e)^\top P_t^e(x_{t+1}^* - \beta_{t+1}^e) - \frac{1}{2}(x_{t+1}^* - \beta_{t+1}^e)^\top P_{t+1}^e(x_{t+1}^* - \beta_{t+1}^e)}_{\text{Part 2}}
\end{aligned}
$$

Part 1 can be bounded by the following

$$
\begin{aligned}
\text{Part 1} &= \frac{1}{2}(x_{t+1}^* - \beta_t^e + x_{t+1}^* - \beta_{t+1}^e)^\top P_t^e(x_{t+1}^* - \beta_t^e - (x_{t+1}^* - \beta_{t+1}^e)) \\
&\le \frac{1}{2}\|x_{t+1}^* - \beta_t^e + x_{t+1}^* - \beta_{t+1}^e\|_2 \|P_t^e\|_2 \|\beta_{t+1}^e - \beta_t^e\|_2 \\
&\le (\bar{x} + \bar{\beta})\upsilon_{max}(\bar{P})\|\beta_{t+1}^e - \beta_t^e\|_2
\end{aligned}
$$

where the last inequality is by Lemma 7, 8 9.

Part 2 can be bounded by the following when $0 \le t \le N - 1$,

$$
\begin{aligned}
\text{Part 2} &= \frac{1}{2}(x_{t+1}^* - \beta_{t+1}^e)^\top (P_t^e - P_{t+1}^e)(x_{t+1}^* - \beta_{t+1}^e) \\
&\le \frac{1}{2}\|x_{t+1}^* - \beta_{t+1}^e\|_2^2 \|P_t^e - P_{t+1}^e\|_2 \le \frac{1}{2}(\bar{x} + \bar{\beta})^2 \|P_t^e - P_{t+1}^e\|_2
\end{aligned}
$$

Therefore, we have

$$
\begin{aligned}
\sum_{t=0}^{N}(h_{t-1}^e(x_t^*) - h_t^e(x_t^*)) &\le \sum_{t=0}^{N-1}(h_t^e(x_{t+1}^*) - h_{t+1}^e(x_{t+1}^*)) \\
&= O\left(\sum_{t=0}^{N-1}(\|\beta_{t+1}^e - \beta_t^e\|_2 + \|P_t^e - P_{t+1}^e\|_2)\right) \quad (19)
\end{aligned}
$$

where the first inequality is by $h_0^e(x) \ge 0$ and $h_{-1}^e(x) = 0$. Thus, by Theorem 2, we have

$$
\text{Regret}(RHTM) = O\left(\zeta^2 (\frac{\sqrt{\zeta} - 1}{\sqrt{\zeta}})^{2K}\left(\sum_{t=1}^{N}(\|P_t^e - P_{t-1}^e\| + \|\beta_t^e - \beta_{t-1}^e\|) + \sum_{t=0}^{N}\|x_{t-1}^e - x_t^e\|\right)\right)
$$

$\square$

### E.2 Proof of Corollary 1

Roughly speaking, the proof is mostly by applying Corollary 2 and by showing $\|\beta_t^e - \beta_{t-1}^e\|$ and $\|x_t^e - x_{t-1}^e\|$ can be bounded by $\|\theta_t - \theta_{t-1}\|$ up to some constants and $\|P_t^e - P_{t-1}^e\| = 0$ in the LQT problem (9) where $Q$ and $R$ are not changing. However, directly applying the results in Theorem 2 and Corollary 2 will result in some extra constant terms because some inequalities used to derive the bounds in Theorem 2 and Corollary 2 are not necessary when $Q, R$ are not changing. Therefore, we will need some intermediate results in the proofs of Theorem 2 and Corollary 2 to prove Corollary 1.

Firstly, by Lemma 2 and Lemma 3, we have

$$
J(FOSS) - J^* = J(FOSS) - \sum_{t=0}^{N-1}\lambda_t^e + \sum_{t=0}^{N-1}\lambda_t^e - J^*
$$

$$\leq c_1 \underbrace{\sum_{t=0}^{N-1} \|x_{t-1}^e - x_t^e\|}_{\text{Part I}} + \underbrace{\sum_{t=0}^{N-1} (h_t^e(x_{t+1}^*) - h_{t+1}^e(x_{t+1}^*))}_{\text{Part II}} + \underbrace{f_N(x_N(0)) - h_0^e(x_0)}_{\text{Part III}}$$

We are going to bound each part by $\sum_t \|\theta_t - \theta_{t-1}\|$ in the following.

**Part I:** We will bound Part I by $\sum_t \|\theta_t - \theta_{t-1}\|$ through showing that $x_t^e = F_1 F_2 \theta_t$ for some matrices $F_1, F_2$. The representation of $x_t^e$ relies on Lemma 5.

By Lemma 5, any steady state $(x, u)$ can be represented as a matrix multiplied by $z$:

$$x = (\underbrace{z^1, \ldots, z^1}_{p_1}, \underbrace{z^2, \ldots, z^2}_{p_2}, \ldots, \underbrace{z^m, \ldots, z^m}_{p_m})^\top =: F_1 z$$

$$u = (z^1, \ldots, z^m)^\top - A(\mathcal{I}, :)x = (I_m - A(\mathcal{I}, :)F_1)z$$

where $F_1 \in \mathbb{R}^{n,m}$ is a binary matrix with full column rank.

Consider cost function $\frac{1}{2}(x - \theta)^\top Q(x - \theta) + \frac{1}{2}u^\top Ru$. By the steady-state representation above, the optimal steady state can be solved by the following unconstrained optimization:

$$\min_z (F_1 z - \theta)^\top Q(F_1 z - \theta) + z^\top (I - A(\mathcal{I}, :)F_1)^\top R(I - A(\mathcal{I}, :)F_1)z$$

Since $F_1$ is full column rank, the function is strongly convex and has the unique solution

$$z^e = F_2 \theta \tag{20}$$

where $F_2 = (F_1^\top Q F_1 + (I - A(\mathcal{I}, :)F_1)^\top R(I - A(\mathcal{I}, :)F_1))^{-1} F_1^\top Q$. Accordingly, the optimal steady state can be represented as

$$x^e = F_1 F_2 \theta, \qquad u^e = (I_m - A(\mathcal{I}, :)F_1)F_2 \theta. \tag{21}$$

Consequently, when $1 \leq t \leq N-1$, $\|x_t^e - x_{t-1}^e\| \leq \|F_1 F_2\|\|\theta_t - \theta_{t-1}\|$. When $t = 0$, $\|x_0^e - x_{-1}^e\| \leq \|F_1 F_2\|\|\theta_0 - \theta_{-1}\|$ holds since $x_{-1}^e = x_0 = \theta_{-1} = 0$. Combining the upper bounds above, we have

$$\text{Part I} = O\left(\sum_{t=0}^{N-1} \|\theta_t - \theta_{t-1}\|\right)$$

**Part II:** By (19) in the proof of Corollary 2, and by noticing that $P_t^e = P^e(Q, R)$ does not change, we have

$$\sum_{t=0}^{N-1} (h_t^e(x_{t+1}^*) - h_{t+1}^e(x_{t+1}^*)) = O\left(\sum_{t=0}^{N-1} \|\beta_{t+1}^e - \beta_t^e\|\right)$$

By Lemma 6, $\beta_t^e = F(Q, R)\theta_t$ for $0 \leq t \leq N - 1$. In addition, since $\beta_N^e = \theta_N = 0$ as defined in (9) and Corollary 2, we can also write $\beta_N^e = F(Q, R)\theta_N$. Thus,

$$\text{Part II} = O\left(\sum_{t=0}^{N-1} \|\beta_{t+1}^e - \beta_t^e\|\right) = O\left(\sum_{t=1}^{N} \|\theta_t - \theta_{t-1}\|\right)$$

**Part III:** By our condition for the terminal cost function, we have $f_N(x_N(0)) = \frac{1}{2}(x_N(0) - \beta_N^e)^\top P^e(x_N(0) - \beta_N^e)$. By Lemma 6, we have $h_0^e(x_0) = \frac{1}{2}(x_0 - \beta_0^e)^\top P^e(x_0 - \beta_0^e)$. So Part III can be bounded by

$$\begin{aligned}
\text{Part III} &= \frac{1}{2}(x_N(0) - \beta_N^e)^\top P^e(x_N(0) - \beta_N^e) - \frac{1}{2}(x_0 - \beta_0^e)^\top P^e(x_0 - \beta_0^e) \\
&= \frac{1}{2}(x_N(0) - \beta_N^e + x_0 - \beta_0^e)^\top P^e(x_N(0) - \beta_N^e - (x_0 - \beta_0^e)) \\
&\leq \frac{1}{2}\|x_N(0) - \beta_N^e + x_0 - \beta_0^e\|\|P^e\|\|x_N(0) - \beta_N^e - (x_0 - \beta_0^e)\| \\
&\leq \frac{1}{2}(\sqrt{n}\bar{x}^e + \bar{\beta} + \bar{\beta})\|P^e\|(\|x_N(0) - x_0\| + \|\beta_N^e - \beta_0^e\|)
\end{aligned}$$

where the last inequality is by $x_0 = 0$, Lemma 4, Lemma 8.

Next we will bound $\|x_N(0) - x_0\|$ and $\|\beta_N^e - \beta_0^e\|$ respectively. Firstly, by $\beta_t^e = F(Q, R)\theta_t$ in Lemma 6, we have

$$\|\beta_N^e - \beta_0^e\| \leq \sum_{t=0}^{N-1} \|\beta_{t+1}^e - \beta_t^e\| \leq \|F(Q, R)\| \sum_{t=0}^{N-1} \|\theta_{t+1} - \theta_t\|$$

Secondly, we will bound $\|x_N(0) - x_0\|$.

$$\|x_N(0) - x_0\| \leq \|x_N(0) - x_{N-1}^e\| + \|x_{N-1}^e - x_0\|$$

$$\leq \|x_N(0) - x_{N-1}^e\| + \sum_{t=0}^{N-1} \|x_t^e - x_{t-1}^e\|$$

$$\leq \|x_N(0) - x_{N-1}^e\| + \|F_1 F_2\| \sum_{t=0}^{N-1} \|\theta_t - \theta_{t-1}\|$$

where the second inequality is by $x_0^e = x_0$, the third inequality is by (21).

Next, we will focus on $\|x_N(0) - x_{N-1}^e\|$. By Lemma 5,

$$x_N(0) = (z_{N-p_1}^{e,1}, \dots, z_{N-1}^{e,1}, z_{N-p_2}^{e,2}, \dots, z_{N-1}^{e,2}, \dots, z_{N-p_m}^{e,m}, \dots, z_{N-1}^{e,m})^\top$$

$$x_{N-1}^e = (z_{N-1}^{e,1}, \dots, z_{N-1}^{e,1}, z_{N-1}^{e,2}, \dots, z_{N-1}^{e,2}, \dots, z_{N-1}^{e,m}, \dots, z_{N-1}^{e,m})^\top$$

As a result,

$$\|x_N(0) - x_{N-1}^e\|^2 \leq \|z_{N-2}^e - z_{N-1}^e\|^2 + \cdots + \|z_{N-p}^e - z_{N-1}^e\|^2$$

$$= \|F_2\|^2 (\|\theta_{N-2} - \theta_{N-1}\|^2 + \cdots + \|\theta_{N-p} - \theta_{N-1}\|^2)$$

where the equality is by (20). Taking square root on both sides yields

$$\|x_N(0) - x_{N-1}^e\| \leq \|F_1\| \sqrt{\|\theta_{N-2} - \theta_{N-1}\|^2 + \cdots + \|\theta_{N-p} - \theta_{N-1}\|^2}$$

$$\leq \|F_2\| (\|\theta_{N-2} - \theta_{N-1}\| + \cdots + \|\theta_{N-p} - \theta_{N-1}\|)$$

$$\leq \|F_2\| (p-1) \sum_{t=N-p}^{N-2} \|\theta_{t+1} - \theta_t\|$$

Combining the bounds above leads to

$$\text{Part III} = O\left(\sum_{t=0}^{N-1} \|\theta_{t+1} - \theta_t\|\right)$$

The proof is completed by summing up the bounds for Part I, II, III.

## F    Proof of Theorem 3

*Proof intuition:* By the problem transformation in Section 3.1, the fundamental limit of the online control problem is equivalent to the fundamental limit of the online convex optimization problem with objective $C(\mathbf{z})$. Therefore, we will focus on $C(\mathbf{z})$. Since the lower bound is for the worst case scenario, we only need to construct some tracking trajectories $\{\theta_t\}$ for Theorem 3 to hold. However, it is generally difficult to construct the tracking trajectories, so we consider randomly generated $\theta_t$ and show that the regret in expectation can be lower bounded. Then, there must exist some realization of the randomly generated $\{\theta_t\}$ such that the regret lower bound holds.

*Formal proof:*

**Step 1: construct LQ tracking.** For simplicity, we construct a single-input system with $n = p$ and $A \in \mathbb{R}^{n,n}$ and $B \in \mathbb{R}^{n \times 1}$ as follows: [6]

$$A = \begin{pmatrix} 0 & 1 & \cdots & 0 \\ \vdots & \ddots & \ddots & \\ & & 0 & 1 \\ 1 & 0 & \cdots & 0 \end{pmatrix}, \quad B = \begin{pmatrix} 0 \\ \vdots \\ 0 \\ 1 \end{pmatrix}$$

$(A, B)$ is controllable because $(B, AB, \ldots, A^{p-1}B)$ is full rank. $A$'s controllability index is $p = n$.

Next, we construct $Q$ and $R$. For any $\zeta > 1$ and $p$, define $\delta = \frac{4}{(\zeta-1)p}$. Let $Q = \delta I_n$ and $R = 1$ for $0 \le t \le N - 1$. Let $P^e = P^e(Q, R)$ be the solution to the DARE. The next lemma shows that $P^e$ is a diagonal matrix and its diagonal entries can be characterized.

**Lemma 10** (Form of $P^e$). *Let $P^e$ denote the solution to the DARE determined by $A, B, Q, R$ defined above. Then $P^e$ satisfies the form*

$$
P^e = \begin{pmatrix} q_1 & 0 & \cdots & 0 \\ 0 & q_2 & \cdots & 0 \\ & & \ddots & \\ 0 & & \cdots & q_n \end{pmatrix},
$$

*where $q_i = q_1 + (i-1)\delta$ for $1 \le i \le n$ and $\delta < q_1 < \delta + 1$.*

*Proof of Lemma 10.* The DARE exists a unique positive definite solution [59]. Suppose the solution is diagonal and substitute it in the DARE as follows.

$$
P^e = Q + A^\top (P^e - P^e B(B^\top P^e B + R)^{-1} B^\top P^e) A
$$

$$
\begin{pmatrix} q_1 & 0 & \cdots & 0 \\ 0 & q_2 & \cdots & 0 \\ & & \ddots & \\ 0 & & \cdots & q_n \end{pmatrix} = \begin{pmatrix} q_n/(1+q_n) + \delta & 0 & \cdots & 0 \\ 0 & q_1 + \delta & \cdots & 0 \\ & & \ddots & \\ 0 & & \cdots & q_{n-1} + \delta \end{pmatrix}
$$

So we have $q_i = q_{i-1} + \delta$ for $1 \le i \le n - 1$, and $q_n/(1+q_n) + \delta = q_1 = q_n - (n-1)\delta$. Thus, $q_n = \frac{n\delta + \sqrt{n^2\delta^2 + 4n\delta}}{2} > n\delta$. It is straightforward that $q_1 = q_n - (n-1)\delta > \delta > 0$, and $q_1 < \delta + 1$ by $q_n/(1+q_n) < 1$. So we have found the unique positive definite solution to the DARE. $\square$

Next, we will construct $\theta_t$. Let $\theta_0 = \theta_N = \beta_N^e = 0$ for simplicity. For $\theta_t$ when $1 \le t \le N - 1$, we divide the $N - 1$ stages into $E$ epochs, each with length $\Delta = \lceil \frac{N-1}{\lfloor \frac{L_N}{2\bar\theta} \rfloor} \rceil$, possibly except the last epoch. This is possible because $1 \le \Delta \le N - 1$ by the conditions in Theorem 3. Thus, $E = \lceil \frac{N-1}{\Delta} \rceil$. Let $\mathcal{J}$ be the first stage of the each epoch: $\mathcal{J} = \{1, \Delta + 1, \ldots, (E-1)\Delta + 1\}$. Let $\theta_t$ for $t \in \mathcal{J}$ independently and identically follow the distribution below.

$$
\Pr(\theta_t^i = a) = \begin{cases} 1/2 & \text{if } a = \sigma \\ 1/2 & \text{if } a = -\sigma \end{cases}, \quad \text{i.i.d. for all } i \in [n], t \in \mathcal{J},
$$

where $\sigma = \frac{\bar\theta}{\sqrt{n}}$. It can be easily verified that $\|\theta\| = \bar\theta$ for any realization of this distribution, so Assumption 3 is satisfied. Let the other $\theta_t$ in each epoch be equal to the $\theta$ at the start of their corresponding epochs, i.e. $\theta_{k\Delta+1} = \theta_{k\Delta+2} = \cdots = \theta_{(k+1)\Delta}$, when $k \le E - 1$, and $\theta_{k\Delta+1} = \cdots = \theta_{N-1}$ when $k = E$. The following inequalities show that the constructed $\{\theta_t\}$ satisfies the variation budget:

$$
\sum_{t=0}^{N} \|\theta_t - \theta_{t-1}\| = \|\theta_1 - \theta_0\| + \sum_{k=1}^{E-1} \|\theta_{k\Delta+1} - \theta_{k\Delta}\| + \|\theta_{N-1} - \theta_N\|
$$

$$
\le \bar\theta + 2(E-1)\bar\theta + \bar\theta = 2\bar\theta E
$$

$$
\le 2\bar\theta \lfloor \frac{L_N}{2\bar\theta} \rfloor \le 2\bar\theta \frac{L_N}{2\bar\theta} = L_N
$$

where the first equality is by $\theta_0 = \theta_{-1} = \theta_N = 0$, the first inequality is by $\|\theta_t\| = \bar\theta$ when $1 \le t \le N - 1$, the second inequality is by $\Delta = \lceil \frac{N-1}{\lfloor \frac{L_N}{2\bar\theta} \rfloor} \rceil \ge \frac{N-1}{\lfloor \frac{L_N}{2\bar\theta} \rfloor}$, and thus $\lfloor \frac{L_N}{2\bar\theta} \rfloor \ge \lceil \frac{N-1}{\Delta} \rceil = E$.

The total cost of our constructed LQ tracking problem is

$$
J(\mathbf{x}, \mathbf{u}) = \sum_{t=0}^{N-1} (\frac{\delta}{2}\|x_t - \theta_t\|^2 + \frac{1}{2}u_t^2) + \frac{1}{2}x_N^\top P^e x_N
$$

We will verify that $C(\mathbf{z})$'s condition number is $\zeta$ in Step 2.

**Step 2: problem transformation and the optimal solution $\mathbf{z}^*$.** By the problem transformation in Section 3.1, we let $z_t = x_t^n$, and the equivalent cost function $C(\mathbf{z})$ is given below.

$$C(\mathbf{z}) = \sum_{t=0}^{N-1} \left( \frac{\delta}{2} \sum_{i=1}^{n} (z_{t-n+i} - \theta_t^i)^2 + \frac{1}{2}(z_{t+1} - z_{t-n+1})^2 \right) + \frac{1}{2} \sum_{i=1}^{n} q_i z_{N-n+i}^2$$

and $z_t = 0$ and $\theta_t = 0$ for $t \le 0$.

Since $C(\mathbf{z})$ is strongly convex, $\min C(\mathbf{z})$ admits a unique optimal solution, denoted as $\mathbf{z}^*$, which is determined by the first-order optimality condition: $\nabla C(\mathbf{z}^*) = 0$. In addition, our constructed $C(\mathbf{z})$ is a quadratic function, so there exists a matrix $H \in \mathbb{R}^{N \times N}$ and a vector $\eta \in \mathbb{R}^N$ such that $\nabla C(\mathbf{z}^*) = H\mathbf{z}^* - \eta = 0$. By the partial gradients of $C(\mathbf{z})$ below,

$$\frac{\partial C}{\partial z_t} = \delta(z_t - \theta_t^n + z_t - \theta_{t+1}^{n-1} + \cdots + z_t - \theta_{t+n-1}^1) + z_t - z_{t+n} + z_t - z_{t-n}, \ 1 \le t \le N-n$$

$$\frac{\partial C}{\partial z_t} = \delta(z_t - \theta_t^n + \cdots + z_t - \theta_{N-1}^{n+t-N+1}) + q_{n+t-N}z_t + z_t - z_{t-n}, \qquad N-n+1 \le t \le N$$

For simplicity and without loss of generality, we assume that $N/n$ is an integer. Then, by Lemma 10, $H$ can be represented as the block matrix below

$$H = \begin{pmatrix} (\delta n + 2)I_n & -I_n & \cdots & \\ -I_n & (\delta n + 2)I_n & \ddots & \\ & \ddots & \ddots & -I_n \\ & & -I_n & (q_n + 1)I_n \end{pmatrix} \in \mathbb{R}^{N \times N}.$$

$\eta$ is a linear combination of $\theta$: for $1 \le t \le N$, we have $\eta_t = \delta(\theta_t^n + \cdots + \theta_{t+n-1}^1) = \delta(e_n^\top \theta_t + \cdots + e_1^\top \theta_{t+n-1})$ where $e_1, \ldots, e_n \in \mathbb{R}^n$ are standard basis vectors and $\theta_t = 0$ for $t \ge N$.

By Gergoskin's Disc Theorem and Lemma 10, $H$'s condition number is $(\delta n + 4)/\delta n = \zeta$ by our choice of $\delta$ in Step 1 and $p = n$. Thus we have shown that $C(\mathbf{z})$'s condition number is $\zeta$.

Since $H$ is strictly diagonally dominant with positive diagonal entries and nonpositive off-diagonal entries, $H$ is invertible and its inverse, denoted by $Y$, is nonnegative. Consequently, the optimal solution can be represented as $\mathbf{z}^* = Y\eta$. Since $\eta$ is linear in $\{\theta_t\}$, $z_t^*$ is also linear in $\{\theta_t\}$ and can be characterized by the following.

$$\begin{aligned} z_{t+1}^* &= \sum_{i=1}^{N} Y_{t+1,i}\eta_i = \delta \sum_{i=1}^{N} Y_{t+1,i} \sum_{j=0}^{n-1} e_{n-j}^\top \theta_{i+j} \\ &= \delta \sum_{k=1}^{N-1} \left( \sum_{i=1}^{n} Y_{t+1,i+k-n} e_i^\top \right) \theta_k \\ &=: \delta \sum_{k=1}^{N-1} v_{t+1,k}\theta_k \end{aligned} \tag{22}$$

where $\theta_t = 0$ for $t \ge N$, $Y_{t+1,i} = 0$ for $i \le 0$, and $v_{t+1,k} := \sum_{i=1}^{n} Y_{t+1,i+k-n}e_i^\top$.

In addition, we are able to show in the next lemma that $Y$ has decaying row entries starting at the diagonal entries. The proof is technical and deferred to the Appendix F.1.

**Lemma 11.** *When $N/n$ is an integer, the inverse of $H$, denoted by $Y$, can be represented as a block matrix*

$$Y = \begin{pmatrix} y_{1,1}I_n & y_{1,2}I_n & \cdots & y_{1,N/n}I_n \\ y_{2,1}I_n & y_{2,2}I_n & \cdots & y_{2,N/n}I_n \\ \vdots & \ddots & \ddots & \vdots \\ y_{N/n,1}I_n & y_{N/n,2}I_n & \cdots & y_{N/n,N/n}I_n \end{pmatrix}$$

*where $y_{t,t+\tau} \ge \frac{1-\rho}{\delta n + 2}\rho^\tau > 0$ for $\tau \ge 0$ and $\rho = \frac{\sqrt{\zeta}-1}{\sqrt{\zeta}+1}$.*

**Step 3: characterize $z_{t+1}(\mathcal{A}^z)$.** For any online control algorithm $\mathcal{A}$, we can define an equivalent online algorithm for $z$, denoted as $\mathcal{A}^z$. $\mathcal{A}^z$, at each time $t$, outputs $z_{t+1}(\mathcal{A}^z)$ based on the predictions and the history, i.e.,

$$z_{t+1}(\mathcal{A}^z) = \mathcal{A}^z(\{\theta_s\}_{s=0}^{t+W-1}), \quad t \geq 0$$

For simplicity, we consider online deterministic algorithm.[7] Notice that $z_{t+1}$ is a random variable because $\theta_1, \ldots, \theta_{t+W-1}$ are random. Based on this observation and Lemma 11, we are able to provide a regret lower bound in Step 4.

**Step 4: prove the regret lower bound on $\mathcal{A}$.** Roughly speaking, the regret occurs when something unexpected happens beyond the prediction window, that is, at each $t$, the prediction window goes as far as $t + W - 1$, but if $\theta_{t+W}$ changes from $\theta_{t+W-1}$, the online algorithm cannot prepare for it, resulting in poor control and positive regret. By our construction, when $t + W \in \mathcal{J}$, $\theta_{t+W}$ changes from $\theta_{t+W-1}$. To study such $t$, we define a set $\mathcal{J}_1 = \{0 \leq t \leq N - W - 1 \mid t + W \in \mathcal{J}\}$. It can be shown that the cardinality of $\mathcal{J}_1$ can be lower bounded by $L_N$ up to some constants:

$$|\mathcal{J}_1| \geq \frac{1}{18\bar{\theta}} L_N \tag{23}$$

The proof of (23) is provided below.

$$
\begin{aligned}
|\mathcal{J}_1| &= |\{W \leq t \leq N - 1 \mid t \in \mathcal{J}\}| \\
&= |\mathcal{J}| - |\{1 \leq t \leq W - 1 \mid t \in \mathcal{J}\}| \\
&= \lceil \frac{N-1}{\Delta} \rceil - \lceil \frac{W-1}{\Delta} \rceil \\
&\geq \lfloor \frac{N-W}{\Delta} \rfloor \\
&\geq \frac{1}{2} \frac{N-W}{\Delta} \\
&\geq \frac{1}{2} \frac{N-W}{N-1+\lfloor \frac{L_N}{2\bar{\theta}} \rfloor} \lfloor \frac{L_N}{2\bar{\theta}} \rfloor \\
&\geq \frac{1}{2} \frac{N - \frac{1}{3}N}{N-1+N+1/2} \lfloor \frac{L_N}{2\bar{\theta}} \rfloor \geq \frac{1}{6} \lfloor \frac{L_N}{2\bar{\theta}} \rfloor \\
&\geq \frac{1}{6} \frac{2}{3} \frac{L_N}{2\bar{\theta}} = \frac{1}{18} \frac{L_N}{\bar{\theta}}
\end{aligned}
$$

where the first inequality is by the definition of the ceiling and floor operators, the second inequality is by $\frac{N-W}{\Delta} \geq 1$ under the conditions on $N, W, L_N$ in Theorem 3, the third inequality is by $\Delta = \lceil \frac{N-1}{\lfloor \frac{L_N}{2\bar{\theta}} \rfloor} \rceil \leq \frac{N-1}{\lfloor \frac{L_N}{2\bar{\theta}} \rfloor} + 1$, the fourth inequality is by $L_N \leq (2N+1)\bar{\theta}$ in Theorem 3's statement, the last inequality is by $L_N \geq 4\bar{\theta}$ in Theorem 3's statement.

Moreover, we can show in Lemma 12 that, for all $t \in \mathcal{J}_1$, the online decision $z_{t+1}(\mathcal{A}^z)$ is different from the optimal solution $z_{t+1}^*$ and the difference is lower bounded,

**Lemma 12.** *For any online algorithm $\mathcal{A}^z$, when $t \in \mathcal{J}_1$,*

$$\mathbb{E} |z_{t+1}(\mathcal{A}^z) - z_{t+1}^*|^2 \geq c_{10} \sigma^2 \rho^{2K}$$

*where $c_{10}$ is a constant determined by $A, B, n, Q, R$ constructed above and $\rho = \frac{\sqrt{\zeta}-1}{\sqrt{\zeta}+1}$.*

The proof is provided in Appendix F.2.

The lower bound on the difference between the online decision and the optimal decision results in a lower bound on the regret. By the $n\delta$-strong convexity of $C(\mathbf{z})$,

$$
\begin{aligned}
\mathbb{E}(C(\mathbf{z}(\mathcal{A}^z)) - C(\mathbf{z}^*)) &\geq \frac{\delta n}{2} \sum_{t \in \mathcal{J}_1} \mathbb{E} |z_{t+1}(\mathcal{A}^z) - z_{t+1}^*|^2 \\
&\geq |\mathcal{J}_1| c_{10} \sigma^2 \rho^{2K}
\end{aligned}
$$

$$\geq \frac{L_N}{18\theta}c_{10}\sigma^2\rho^{2K} = \Omega(L_N\rho^{2K})$$

By the equivalence between $\mathcal{A}$ and $\mathcal{A}^z$, we have $\mathbb{E}\,J(\mathcal{A}) - \mathbb{E}\,J^* = \Omega(\rho^{2K}L_N)$. By the property of expectation, there must exist some realization of the random $\{\theta_t\}$ such that $J(\mathcal{A}) - J^* = \Omega(\rho^{2K}L_N)$, where $\rho = \frac{\sqrt{\zeta}-1}{\sqrt{\zeta}+1}$. This completes the proof. $\qquad\square$

### F.1   Proof of Lemma 11

*Proof.* Since $H$ is a block matrix

$$H = \begin{pmatrix} (\delta n + 2)I_n & -I_n & \cdots & \\ -I_n & (\delta n + 2)I_n & \ddots & \\ & \ddots & \ddots & -I_n \\ & & -I_n & (q_n+1)I_n \end{pmatrix}$$

its inverse matrix $Y$ can also be represented as a block matrix. Moreover, let

$$H_1 = \begin{pmatrix} \delta n + 2 & -1 & \cdots & 0 \\ -1 & \delta n + 2 & \ddots & 0 \\ \vdots & \ddots & \ddots & \vdots \\ 0 & \cdots & -1 & q_n + 1 \end{pmatrix}$$

and define $\bar{Y} = (H_1)^{-1} = (y_{ij})_{i,j=1}^{N/n}$. Then the inverse matrix $Y$ can be represented as the block matrix: $Y = (y_{ij}I_n)_{i,j=1}^{N/n}$.

Now, it suffices to provide a lower bound on $y_{ij}$.

Since $H_1$ is a symmetric positive definite tridiagonal matrix, by [63], the inverse has an explicit formula given by $(H_1)_{ij}^{-1} = a_i b_j$ and

$$a_t = \frac{\rho}{1-\rho^2}\left(\frac{1}{\rho^t} - \rho^t\right)$$

$$b_t = c_3\frac{1}{\rho^{N-t}} + c_4\rho^{N-t}$$

$$c_3 = b_N\left(\frac{(q_n+1)\rho - \rho^2}{1-\rho^2}\right)$$

$$c_4 = b_N\frac{1 - (q_n+1)\rho}{1-\rho^2}$$

$$b_N = \frac{1}{-a_{N-1} + (q_n+1)a_N}$$

In the following, we will show $y_{t,t+\tau} = a_t b_{t+\tau} \geq \frac{1-\rho}{\delta n+2}\rho^\tau$ when $\tau \geq 0$. Firstly, it is easy to verify that

$$\rho^t a_t = \frac{\rho}{1-\rho^2}(1-\rho^{2t}) \geq \rho$$

since $t \geq 1$ and $\rho < 1$.

Secondly, we bound $b_N$ in the following way:

$$\rho^{-N}b_N = \frac{1}{(q_n+1)(1-\rho^{2N}) - (\rho - \rho^{2N-1})}\frac{1-\rho^2}{\rho} \geq \frac{1}{(\delta n+2)}\frac{1-\rho^2}{\rho}$$

because $0 < (q_n+1)(1-\rho^{2N}) - (\rho - \rho^{2N-1}) \leq (\delta n + 2)$ by $n\delta < q_n < n\delta + 1$ in Lemma 10.

Thirdly, we bound $b_{t+\tau}$. When $1 - (q_n+1)\rho \geq 0$

$$\rho^{N-t-\tau}b_{t+\tau} = b_N\left(\frac{(q_n+1)\rho - \rho^2}{1-\rho^2}\right) + b_N\frac{1 - (q_n+1)\rho}{1-\rho^2}\rho^{2(N-t-\tau)}$$

$$\geq b_N \left( \frac{(q_n + 1)\rho - \rho^2}{1 - \rho^2} \right)$$

$$\geq b_N \left( \frac{(\delta n + 1)\rho - \rho^2}{1 - \rho^2} \right)$$

$$= \frac{1 - \rho}{1 - \rho^2} b_N$$

where the first inequality is by $1 - (q_n + 1)\rho \geq 0$, the second inequality is by $qn > n\delta$ in Lemma 10, and the last equality is by $\rho^2 - (\delta n + 2)\rho + 1 = 0$.

When $1 - (q_n + 1)\rho < 0$

$$\rho^{N-t-\tau} b_{t+\tau} = b_N \left( \frac{(q_n + 1)\rho - \rho^2}{1 - \rho^2} \right) + b_N \frac{1 - (q_n + 1)\rho}{1 - \rho^2} \rho^{2(N-t-\tau)}$$

$$\geq b_N \left( \frac{(q_n + 1)\rho - \rho^2}{1 - \rho^2} \right) + b_N \frac{1 - (q_n + 1)\rho}{1 - \rho^2}$$

$$\geq b_N \geq \frac{1 - \rho}{1 - \rho^2} b_N$$

where the first inequality is by $1 - (q_n + 1)\rho < 0, \rho \leq 1$, the second inequality is by $\rho^{2(N-t-\tau)} \leq 1$. Thus, we obtained a lower bound for $b_{t+\tau}$.

Combining bounds of $a_t, b_{t+\tau}, b_N$ together yields

$$y_{t,t+\tau} = a_t b_{t+\tau} \geq \rho b_N \frac{1 - \rho}{1 - \rho^2} \rho^{\tau - N} \geq \frac{1 - \rho}{(\delta n + 2)} \rho^\tau$$

$\square$

### F.2 Proof of Lemma 12

*Proof.* By our construction, $\theta_t$ is random, $z_{t+1}^{\mathcal{A}}$ is also random and its randomness is provided by $\theta_1, \ldots, \theta_{t+W-1}$, while $z_{t+1}^*$ is determined by all $\theta_t$. When $t \in \mathcal{J}_1$,

$$\mathbb{E} \, |z_{t+1}^{\mathcal{A}} - z_{t+1}^*|^2 = \mathbb{E} \, |z_{t+1}^{\mathcal{A}} - \delta \sum_{i=1}^{N-1} v_{t+1,i}\theta_i|^2$$

$$= \mathbb{E} \, |z_{t+1}^{\mathcal{A}} - \delta \sum_{i=1}^{t+W-1} v_{t+1,i}\theta_i\|^2 + \delta^2 \, \mathbb{E} \, | \sum_{i=t+W}^{N-1} v_{t+1,i}\theta_i|^2$$

$$\geq \delta^2 \, \mathbb{E} \, | \sum_{i=t+W}^{N-1} v_{t+1,i}\theta_i|^2,$$

where the first equality is by (22), the second equality is by $\mathbb{E}\, \theta_\tau = 0$ for all $\tau$, and $\theta_{t+W}, \ldots, \theta_N$ are independent of $\theta_1, \ldots, \theta_{t+W-1}$ when $t \in \mathcal{J}_1$.

Further,

$$\mathbb{E} \, | \sum_{i=t+W}^{N-1} v_{t+1,i}\theta_i|^2 = \mathbb{E} \, | \sum_{i=t+W}^{t+W+\Delta-1} v_{t+1,i}\theta_{t+W}|^2 + \cdots + \mathbb{E} \, | \sum_{i=(E-1)\Delta+1}^{N-1} v_{t+1,i}\theta_{(E-1)\Delta+1}|^2$$

$$= \| \sum_{i=t+W}^{t+W+\Delta-1} v_{t+1,i}\|^2 \sigma^2 + \cdots + \| \sum_{i=(E-1)\Delta+1}^{N-1} v_{t+1,i}\|^2 \sigma^2$$

$$\geq \sigma^2 \sum_{i=t+W}^{N-1} \|v_{t+1,i}\|^2$$

$$= \sigma^2 \sum_{i=t+W}^{N-1} (\sum_{k=0}^{n-1} Y_{t+1,i-k}^2) \geq \sigma^2 \sum_{i=t+1+W-n}^{N-1} Y_{t+1,i}^2$$

$$= \sigma^2 \sum_{i=t+1+W-n}^{N} Y_{t+1,i}^2$$

where the first equality is because the theta in one epoch are equal by our construction, the second equality is because $\mathrm{cov}(\theta_\tau) = \sigma^2 I_n$, the first inequality is because the entries of $v_{t+1,i}$ are nonnegative, the third equality is by the definition of $v_{t+1,i}$ in (22), and the last equality is because when $t \in \mathcal{J}_1, Y_{t+1,N} = 0$.

When $1 \leq W \leq n$, $\sum_{i=t+1+W-n}^{N} Y_{t+1,i}^2 \geq Y_{t+1,t+1}^2 = Y_{t+1,t+1+n\lfloor \frac{W-1}{n} \rfloor}^2$. When $W > n$, $\sum_{i=t+1+W-n}^{N} Y_{t+1,i}^2 \geq Y_{t+1,t+1+n\lceil \frac{W-n}{n} \rceil}^2$. Moreover, when $W \geq 1$, $\lceil \frac{W-n}{n} \rceil = \lfloor \frac{W-1}{n} \rfloor$. In summary, for $W \geq 1$,

$$\sum_{i=t+1+W-n}^{N} Y_{t+1,i}^2 \geq Y_{t+1,t+1+n\lfloor \frac{W-1}{n} \rfloor}^2 \geq \rho^{2K} \left( \frac{1-\rho}{\delta n + 2} \right)^2$$

where the last inequality is by Lemma 11. This completes the proof.

$\square$

# G Proofs of the LQT's properties used in Appendix E

In this section, we provide proofs for the properties of LQ tracking (LQT) used in Appendix E.

## G.1 Preliminaries: dynamic programming for finite-horizon LQT

In this section, we consider a discrete time LQ tracking problem with time-varying cost functions and time-invariant dynamical system:

$$\min_{x_t, u_t} \frac{1}{2} \sum_{t=0}^{N-1} \left[ (x_t - \theta_t)^\top Q_t (x_t - \theta_t) + u_t^\top R_t u_t \right] + \frac{1}{2} (x_N - \theta_N)^\top Q_N (x_N - \theta_N)$$

$$\text{s.t.} \quad x_{t+1} = A x_t + B u_t, \qquad t = 0, \ldots, N-1$$

where $x_0 = 0$ for simplicity.

The problem can be solved by dynamic programming.

**Theorem 4** (Dynamic programming for the finite-horizon LQT). *Consider a finite-horizon time-varying LQ tracking problem. Let $V_t(x_t)$ be the cost to go from $k = t$ to $k = N$, then*

$$V_t(x_t) = \frac{1}{2} (x_t - \beta_t)^\top P_t (x_t - \beta_t) + \frac{1}{2} \sum_{k=t}^{N-1} (A\theta_k - \beta_{k+1})^\top H_k (A\theta_k - \beta_{k+1})$$

*for $t = 0, \ldots, N$. The parameters can be obtained by*

$$P_t = Q_t + A^\top M_t A, \quad t = 0, \ldots, N-1, \quad P_N = Q_N$$

$$M_t = P_{t+1} - P_{t+1} B (R_t + B^\top P_{t+1} B)^{-1} B^T P_{t+1}, \quad t = 0, \ldots, N-1$$

$$\beta_t = (Q_t + A^\top M_t A)^{-1} (Q_t \theta_t + A^\top M_t \beta_{t+1}), \quad t = 0, \ldots, N-1$$

$$\beta_N = \theta_N$$

$$H_t = M_t - M_t A (Q_t + A^\top M_t A)^{-1} A^\top M_t, \quad t = 0, \ldots, N-1$$

*The optimal controller is*

$$u_t^* = -K_t x_t + K_t' \beta_{t+1}, \quad t = 0, \ldots, N-1$$

*where the parameters are*

$$K_t = (R_t + B^\top P_{t+1} B)^{-1} B^\top P_{t+1} A$$

$$K_t' = (R_t + B^\top P_{t+1} B)^{-1} B^\top P_{t+1}$$

*There is another way to write the optimal controller:*

$$u_t^* = -K_t x_t + K_t^\alpha \alpha_{t+1} \quad t = 0, \ldots, N-1$$

*where the parameters are*

$$K_t^\alpha = (R_t + B^\top P_{t+1} B)^{-1} B^\top$$
$$\alpha_t = P_t \beta_t$$
$$\alpha_t = Q_t \theta_t + (A - BK_t)^\top \alpha_{t+1}, \quad t = 0, \ldots, N-1$$
$$\alpha_N = P_N \theta_N$$

*Proof.* The proof is straightforward by following dynamic programming procedures.

Firstly, it is direct to verify that $V_N(x_N) = \frac{1}{2}(x_N - \theta_N)^\top Q_N(x_N - \theta_N)$. Then, suppose the claim of Theorem 4 is true at $t+1$, we will verify the stage $t$ in the following.

$$
\begin{aligned}
V_t(x_t) =& \min_{u_t} \Big[ \frac{1}{2}(x_t - \theta_t)^\top Q_t(x_t - \theta_t) + \frac{1}{2} u_t^\top R_t u_t + V_{t+1}(Ax_t + Bu_t) \Big] \\
=& \frac{1}{2} \min_{u_t} \Big[ (x_t - \theta_t)^\top Q_t(x_t - \theta_t) + u_t^\top R_t u_t + (Ax_t + Bu_t - \beta_{t+1})^\top P_{t+1}(Ax_t + Bu_t - \beta_{t+1}) \\
& + \sum_{k=t+1}^{N-1} (A\theta_k - \beta_{k+1})^\top H_k(A\theta_k - \beta_{k+1}) \Big] \\
=& \frac{1}{2}(Ax_t - \beta_{t+1})^\top (P_{t+1} - P_{t+1}B(R_t + B^\top P_{t+1}B)^{-1}B^\top P_{t+1})(Ax_t - \beta_{t+1}) \\
& + \frac{1}{2}(x_t - \theta_t)^\top Q_t(x_t - \theta_t) + \frac{1}{2} \sum_{k=t+1}^{N-1} (A\theta_k - \beta_{k+1})^\top H_k(A\theta_k - \beta_{k+1}) \\
=& \frac{1}{2}(Ax_t - \beta_{t+1})^\top M_t(Ax_t - \beta_{t+1}) + \frac{1}{2}(x_t - \theta_t)^\top Q_t(x_t - \theta_t) \\
& + \frac{1}{2} \sum_{k=t+1}^{N-1} (A\theta_k - \beta_{k+1})^\top H_k(A\theta_k - \beta_{k+1}) \\
=& \frac{1}{2}(x_t - \beta_t)^\top P_t(x_t - \beta_t) - \frac{1}{2}(Q_t \theta_t + A^\top M_t \beta_{t+1})^\top (Q_t + A^\top M_t A)^{-1}(Q_t \theta_t + A^\top M_t \beta_{t+1}) \\
& + \frac{1}{2}\theta_t^\top Q_t \theta_t + \frac{1}{2}\beta_{t+1}^\top M_t \beta_{t+1} + \frac{1}{2} \sum_{k=t+1}^{N-1} (A\theta_k - \beta_{k+1})^\top H_k(A\theta_k - \beta_{k+1}) \\
=& \frac{1}{2}(x_t - \beta_t)^\top P_t(x_t - \beta_t) + \frac{1}{2} \sum_{k=t}^{N-1} (A\theta_k - \beta_{k+1})^\top H_k(A\theta_k - \beta_{k+1})
\end{aligned}
$$

where the third equality is by noticing that the optimal control input is

$$u_t^* = -(R_t + B^\top P_{t+1}B)^{-1}B^\top P_{t+1}(Ax_t - \beta_{t+1}) = -K_t x_t + K_t' \beta_{t+1},$$

the fourth equality is by $M_t$'s definition, the fifth equality is by combining the two quadratic terms of $x_t$ as one quadratic term with a constant, and the last equality is by definition. $\qquad\square$

## G.2 Proof of Lemma 9

In the following, we first prove that the recursive solution $P_t$ to the finite-horizon LQT is bounded. Then, we can prove Lemma 9 by taking limits.

**Lemma 13** (Bounded $P_t$ for finite-horizon LQT). *Consider a finite-horizon time-varying LQT problem. For any $N$, any $0 \le t \le N$, any $Q_t \in \mathcal{Q}, R_t \in \mathcal{R}, Q_N \in \mathcal{P}$, we have $P_t \in \mathcal{P}$ where $P_t$ is defined in Theorem 4.*

*Proof.* In the following, we use the notations and definitions introduced in Appendix E.1 and Theorem 4.

Since $P_t$ does not depend on $\theta_t$, we let $\theta_t = 0$ and consider the LQR problem for simplicity. Since $\underline{Q} \leq Q_t \leq \bar{Q}, \underline{R} \leq R_t \leq \bar{R}$, for $0 \leq t \leq N-1$ and $\underline{P} \leq Q_N \leq \bar{P}$, we have for any $x_t, u_t, k,$ $\bar{Q}_t, R_t, Q_N,$

$$\sum_{t=k}^{N-1} (x_t^\top Q_t x_t + u_t^\top R_t u_t) + x_N^\top Q_N x_N \leq \sum_{t=k}^{N-1} (x_t^\top \bar{Q} x_t + u_t^\top \bar{R} u_t) + x_N^\top \bar{P} x_N$$

$$\sum_{t=k}^{N-1} (x_t^\top Q_t x_t + u_t^\top R_t u_t) + x_N^\top Q_N x_N \geq \sum_{t=k}^{N-1} (x_t^\top \underline{Q} x_t + u_t^\top \underline{R} u_t) + x_N^\top \underline{P} x_N$$

Taking minimum over all feasible trajectories on both sides yields

$$\min \sum_{t=k}^{N-1} (x_t^\top Q_t x_t + u_t^\top R_t u_t) + x_N^\top Q_N x_N \leq \min \sum_{t=k}^{N-1} (x_t^\top \bar{Q} x_t + u_t^\top \bar{R} u_t) + x_N^\top \bar{P} x_N$$

$$\min \sum_{t=k}^{N-1} (x_t^\top Q_t x_t + u_t^\top R_t u_t) + x_N^\top Q_N x_N \geq \min \sum_{t=k}^{N-1} (x_t^\top \underline{Q} x_t + u_t^\top \underline{R} u_t) + x_N^\top \underline{P} x_N$$

Notice that the left-hand-side terms of both inequalities are equal to $x_k^\top P_k x_k$. Moreover, notice that

$$x_k^\top \bar{P} x_k = \min_{x_{t+1} = A x_t + B u_t} \sum_{t=k}^{N-1} (x_t^\top \bar{Q} x_t + u_t^\top \bar{R} u_t) + x_N^\top \bar{P} x_N$$

because $\bar{P} = P^e(\bar{Q}, \bar{R})$ is the solution to the DARE. The same holds for $\underline{P}$. Therefore, we have

$$x_k^\top \underline{P} x_k \leq x_k^\top P_k x_k \leq x_k^\top \bar{P} x_k$$

for any $x_k$. Thus, $\underline{P} \leq P_k \leq \bar{P}$, i.e. $P_k \in \mathcal{P}$. $\qquad \square$

*Proof of Lemma 9.* In the following, we use the notations and definitions introduced in Appendix E.1 and Theorem 4. Since $P^e$ is not influenced by $\theta_t$, we let $\theta_t = 0$ for simplicity. Consider a *finite-horizon* LQR problem: $\sum_{k=0}^{N-1} (x_k^\top Q x_k + u_k^\top R u_k) + x_N^\top Q_N x_N$, where $Q_N \in \mathcal{P}$. By Lemma 13, we have $P_k \in \mathcal{P}$. Since $P_k \to P^e$ as $k \to -\infty$ [59], and since $\mathcal{P}$ is a closed set [64], we have $P^e \in \mathcal{P}$. Since $P^e$ and $\bar{P}$ are positive definite, we have $\|P^e\|_2 \leq v_{max}(\bar{P})$.

$\qquad \square$

### G.3 Proof of Lemma 6

In the following, we will provide and prove an enhanced version of Lemma 6 with detailed characterization of the solution to the Bellman equations in Proposition 1.

**Proposition 1** (Optimal solution to average-cost LQ tracking). *Suppose $(A, B)$ is controllable, $Q, R$ are positive definite. The optimal average cost $\lambda^e$ does not depend on the initial state $x_0$ and is equal to*

$$\lambda^e = \frac{1}{2} (A\theta - \beta^e)^\top H^e (A\theta - \beta^e),$$

*where $M^e = P^e - P^e B(R + B^\top P^e B)^{-1} B^\top P^e$ and $H^e = M^e - M^e A(Q + A^\top M^e A)^{-1} A^\top M^e$.*

*In addition, a bias function of the Bellman equations $h^e(x) + \lambda^e = \min_u(f(x) + g(u) + h^e(Ax + Bu))$ can be represented by*

$$h^e(x) = \frac{1}{2} (x - \beta^e)^\top P^e (x - \beta^e).$$

*where $P^e = P^e(Q, R)$.*

*The optimal controller is*

$$u = -K^e x + K' \beta^e$$

*where $K^e = (R + B^\top P^e B)^{-1} B^\top P^e A$, $K' = (R + B^\top P^e B)^{-1} B^\top P^e$, and $\beta^e$ satisfies*

$$\beta^e = (P^e)^{-1} \alpha^e = F\theta \tag{24}$$

*where $\alpha^e = Q\theta + (A - BK^e)^\top \alpha^e$ and thus $F = (P^e)^{-1}(I - (A - BK^e)^\top)^{-1} Q$.*

*Proof of Proposition 1.* It is easy to see that the formulas of $\lambda^e$, $h^e(x)$, and the optimal controller are the limits of the corresponding formulas or the limiting solutions to the corresponding iterative equations under fixed $Q, R, \theta$ in Theorem 4. However, to formally prove these formulas, we still need to prove the existence of the limits, which is the focus of the following proof. In particular, the proof consists of three parts: i) verify the formula of the optimal average cost $\lambda^e$, ii) verify the formula of the bias function $h^e(x)$, iii) verify the formula of the optimal controller.

*Part i): Verify the formula of $\lambda^e$.* Consider a finite horizon LQT problem:

$$\min_{x_t, u_t} \frac{1}{2} \sum_{t=0}^{N-1} \left[ (x_t - \theta)^\top Q(x_t - \theta) + u_t^\top R u_t \right]$$

$$\text{s.t.} \quad x_{t+1} = Ax_t + Bu_t, \qquad t = 0, \ldots, N-1$$

Given an initial state $x_0$, by Theorem 4, the total optimal cost in $N$ time steps is

$$J_N^*(x_0) = \frac{1}{2}(x_0 - \beta_0)^\top P_0 (x_0 - \beta_0) + \frac{1}{2} \sum_{k=0}^{N-1} (A\theta - \beta_{k+1})^\top H_k (A\theta - \beta_{k+1})$$

If we can show that $\beta_k \to \beta^e$ and $P_k \to P^e$ and $H_k \to H^e$ as $k \to -\infty$, then, consequently, we will have $\frac{1}{2}(A\theta - \beta_{k+1})^\top H_k (A\theta - \beta_{k+1}) \to \frac{1}{2}(A\theta - \beta^e)^\top H^e (A\theta - \beta^e)$ as $k \to -\infty$, and bounded $\frac{1}{2}(x_0 - \beta_0)^\top P_0 (x_0 - \beta_0)$ for fixed $x_0$. Then the formula of the optimal average cost in infinite horizon can be proved by

$$\lambda^e = \lim_{N \to +\infty} \left[ \frac{1}{N} \min_{x_{t+1} = Ax_t + Bu_t} \left( \frac{1}{2} \sum_{t=0}^{N-1} \left( (x_t - \theta)^\top Q(x_t - \theta) + u_t^\top R u_t \right) \right) \right]$$

$$= \lim_{N \to +\infty} \left[ \frac{1}{N} \left( \frac{1}{2}(x_0 - \beta_0)^\top P_0 (x_0 - \beta_0) + \frac{1}{2} \sum_{k=0}^{N-1} (A\theta - \beta_{k+1})^\top H_k (A\theta - \beta_{k+1}) \right) \right]$$

$$= \frac{1}{2}(A\theta - \beta^e)^\top H^e (A\theta - \beta^e),$$

Therefore, it suffices to prove $\beta_k \to \beta^e$, $P_k \to P^e$ and $H_k \to H^e$ as $k \to -\infty$.

By Proposition 4.4.1 [59], $P_k \to P^e$ as $k \to -\infty$. Then, $M_k \to M^e$ as $k \to -\infty$ since $M_k$ is a continuous function of $P_k$ by noticing that the matrix inverse operator is continuous when the matrix is invertible. Similarly, $H_k \to H^e$ as $k \to -\infty$ since $H_k$ is a continuous function of $M_k$. In addition, $K_k \to K^e$, and $K_k^\alpha \to K^\alpha$ and $K_k' \to K'$ as $k \to -\infty$ since $K_k, K_k^\alpha, K_k'$ are continuous functions of $P_k$.

To show $\beta_k \to \beta^e$, we only need to show $\alpha_k \to \alpha^e$ as $k \to -\infty$ since $\beta_k = P_k^{-1}\alpha_k$. $\alpha_k$ satisfies the recursive equation $\alpha_k = Q\theta + (A - BK_{k+1})^\top \alpha_{k+1}$. Since $(A - BK_k)^\top \to (A - BK^e)^\top$ as $k \to -\infty$ and $(A - BK^e)^\top$ is a stable matrix, by the claim below, we can show $\alpha_k \to \alpha^e$ as $k \to -\infty$. Then, the proof of Part i) is completed.

**Claim:** *If $A_t \to A$ and $A$ is stable, then the state of the system $x_{t+1} = A_t x_t + \eta$ will converge to $x^s$, where $x^s = Ax^s + \eta$, for any bounded initial value $x_0$.*

The proof of the claim lemma is provided at the end of this subsection.

*Part ii): Verify $h^e(x)$'s formula.* The proof is by showing the Bellman equations hold under the formulas of $h^e(x)$ and $\lambda^e$ provided in the statement of the lemma.

$$\min_u \left[ \frac{1}{2}(x - \theta)^\top Q(x - \theta) + \frac{1}{2} u^\top R u + \frac{1}{2}(Ax + Bu - \beta^e)^\top P^e (Ax + Bu - \beta^e) \right]$$

$$= \frac{1}{2}(x - \theta)^\top Q(x - \theta) + \frac{1}{2}(Ax - \beta^e)^\top M^e (Ax - \beta^e)$$

$$+ \min_u \frac{1}{2}(u + K^e x - K'\beta^e)^\top (R + B^\top P^e B)(u + K^e x - K'\beta^e)$$

$$= \frac{1}{2}(x - \theta)^\top Q(x - \theta) + \frac{1}{2}(Ax - \beta^e)^\top M^e (Ax - \beta^e)$$

$$= \frac{1}{2}(A\theta - \beta^e)^\top H^e(A\theta - \beta) + \frac{1}{2}(x - \beta^e)^\top P^e(x - \beta^e)$$

where the last equality is by $Q + A^\top M^e A = P^e$, $\beta^e = (P^e)^{-1}\alpha^e$, $\alpha^e = Q\theta + (A - BK^e)^\top \alpha^e$ and the formulas of $K^e, M^e$.

*Part iii): Verify the formula of the optimal controller.* We prove $u = -K^e x + K'\beta^e$ is the optimal controller by showing that the average cost by implementing this controller is no more than the optimal average cost $\lambda^e$. Let $x_t, u_t$ be the state and control at $t$ by implementing the controller $u = -K^e x + K'\beta^e$.

$$\frac{1}{N}\left(\frac{1}{2}\sum_{t=0}^{N-1}\left[(x_t - \theta)^\top Q(x_t - \theta) + u_t^\top R u_t\right]\right)$$

$$\leq \frac{1}{N}\left(\frac{1}{2}\sum_{t=0}^{N-1}\left[(x_t - \theta)^\top Q(x_t - \theta) + u_t^\top R u_t\right] + \frac{1}{2}(x_N - \beta^e)^\top P^e(x_N - \beta^e)\right)$$

$$= \frac{1}{N}\left(\frac{1}{2}(x_0 - \beta^e)^\top P^e(x_0 - \beta^e) + \frac{1}{2}\sum_{k=0}^{N-1}(A\theta - \beta^e)^\top H^e(A\theta - \beta^e)\right)$$

where the last equality is by Theorem 4. Taking $N \to +\infty$ on both sides, we have

$$\lim_{N\to+\infty}\frac{1}{N}\frac{1}{2}\sum_{t=0}^{N-1}\left[(x_t - \theta)^\top Q(x_t - \theta) + u_t^\top R u_t\right] \leq \frac{1}{2}(A\theta - \beta^e)^\top H^e(A\theta - \beta^e) = \lambda^e$$

This completes the proof. $\qquad\square$

**Proof of Claim:** Define the error term $d_t = x_t - x^s$. The dynamics of $d_t$ is $d_{t+1} = Ad_t + w_t$, where $w_t = (A_t - A)(d_t + x^s)$. It suffices to show $d_t \to 0$ as $t \to +\infty$. In the following, we will first prove two facts, based on which we prove $d_t \to 0$.

**Fact 1:** *Consider a stable matrix $A$ and a sequence of uniformly bounded vectors: $\|v_t\|_2 \leq D$ for all $t$. There exists a constant $c_6 > 0$ determined by $A$, such that, for any $k = 1, 2, \ldots,$*

$$\|\sum_{t=0}^{k-1} A^t v_t\|_2 \leq c_6 D.$$

**Proof of Fact 1:** This is a consequence of the fact that exponential stability implies bounded-input-bounded-output stability. To see this, consider a system $x_{t+1} = Ax_t + u_t$ with $x_0 = 0$. Since $A$ is stable, the system is exponentially stable. By Theorem 9.4 [53], the exponential stability implies the bounded-input-bounded-output stability. Thus, there exists $c_6$ such that $\|x_k\|_2 \leq c_6 D$ for any $k$ and any input sequence satisfying $\|u_t\|_2 \leq D$ for all $t$.

For any $k \geq 0$, consider inputs $u_t = v_{k-1-t}$ for $0 \leq t \leq k - 1$, then $x_k = \sum_{t=0}^{k-1} A^t v_t$. Since $\|u_t\| \leq D$, we have $\|x_k\| \leq c_6 D$, which completes the proof. $\qquad\square$

**Fact 2:** *There exists a constant $D > 0$, such that $\max_{t\geq 0}(\|x^s\|, \|d_t\|) \leq D$.*

**Proof of Fact 2:** Since $A_t \to A$, for $\epsilon_1 = 1/(4c_6)$, there exists $N_1$, such that when $t \geq N_1$, $\|A_t - A\| \leq \epsilon_1$. Since $A$ is stable, we have $A^t \to 0$, so for $\epsilon_2 = 1/2$, there exists $N_2$, such that when $t > N_2$, $\|A^t\| \leq \epsilon_2$.

Let $D = \max(\|d_0\|, \ldots, \|d_{N_1+N_2}\|, \|x^s\|)$. By definition, $\|d_t\| \leq D$ for $t \leq N_1 + N_2$. We can show $\|d_{N_1+N_2+1}\| \leq D$ in the following.

$$\|d_{N_1+N_2+1}\| = \|A^{N_2+1}d_{N_1} + w_{N_1+N_2} + Aw_{N_1+N_2-1} + \cdots + A^{N_2}w_{N_1}\|$$

$$\leq \|A^{N_2+1}\|_2 D + \|w_{N_1+N_2} + Aw_{N_1+N_2-1} + \cdots + A^{N_2}w_{N_1}\|$$

$$\leq \epsilon_2 D + c_6 \max_{N_1\leq k\leq N_1+N_2}\|w_k\|$$

$$\leq \epsilon_2 D + 2c_6\epsilon_1 D = (1/2 + 1/2)D = D$$

where the second inequality is by Fact 1 and the definitions of $\epsilon_2$, and the third inequality is by $w_k = (A_k - A)(d_k + x^s)$, $k \geq N_1$, and the definitions of $D$ and $\epsilon_1$.

It can be shown by induction that $\|d_t\| \leq D$ for any $t \geq N_1 + N_2 + 1$ in the same way, which completes the proof. $\qquad\square$

**Prove $d_t \to 0$.** We will show that for any $\epsilon_3 > 0$, there exists $N_3$, such that $\|d_t\|_2 \leq \epsilon_3$ when $t > N_3$. The proof is very similar to the proof of Fact 2.

Since $A_t \to A$, when $\epsilon_1' = \epsilon_3/(4c_6 D)$, there exists $N_1'$, such that when $t \geq N_1'$, $\|A_t - A\| \leq \epsilon_1'$, where $D$ is defined in Fact 2. Since $A$ is stable, we have $A^t \to 0$, so when $\epsilon_2' = \epsilon_3/(2D)$, there exists $N_2'$, such that when $t > N_2'$, $\|A^t\| \leq \epsilon_2'$. Let $N_3 = N_1' + N_2'$. When $t > N_3$,

$$
\begin{aligned}
\|d_{t+1}\| &= \|A^{t-N_1'+1} d_{N_1'} + w_t + A w_{t-1} + \cdots + A^{t-N_1'} w_{N_1'}\| \\
&\leq \|A^{t-N_1'+1}\| D + \|w_t + A w_{t-1} + \cdots + A^{t-N_1'} w_{N_1'}\|_2 \\
&\leq \epsilon_2' D + c_6 \max_{N_1' \leq k \leq t} \|w_k\| \\
&\leq \epsilon_2' D + 2c_6 \epsilon_1' D = (1/2 + 1/2)\epsilon_3 = \epsilon_3
\end{aligned}
$$

where the second inequality is by Fact 1 and the definitions of $\epsilon_2$, and the third inequality is by $w_k = (A_k - A)(d_k + x^s)$, $k \geq N_1$, $\max_{k \geq 0}(\|d_k\|, \|x^s\|) \leq D$, and the definition of $\epsilon_1'$.

This completes the proof of the claim. $\qquad\square$

## G.4 Proof of Lemma 7.

Let $x_t^*$, $u_t^*$ denote the optimal state and the optimal control input at time $t$ respectively. By Theorem 4, the optimal controller is $u_t^* = -K_t x_t^* + K_t^\alpha \alpha_t$. For ease of notation, define

$$
D_t := A - BK_t.
$$

Then, the dynamical system of $x_t^*$ can be represented as

$$
x_{t+1}^* = D_t x_t^* + BK_t^\alpha \alpha_{t+1}.
$$

**Proof outline:** We will prove $x_t^*$ is bounded by three steps: 1) show that system $x_{t+1} = D_t x_t$ is exponentially stable, 2) show that $BK_t^\alpha \alpha_{t+1}$ is bounded, 3) show $x_t^*$ is bounded by the fact that exponentially stable systems are bounded-input-bounded-output stable.

*Step 1: show $x_{t+1} = D_t x_t$ is exponentially stable by a Lyapunov function.*

**Lemma 14** (Exponential stability). *Consider dynamical system $x_{t+1} = D_t x_t$. Define the state transition matrix:*

$$
\Phi(t, t_0) = D_{t-1} \cdots D_{t_0}
$$

*for $t \geq t_0$, and $\Phi(t, t) = I$. For any $N$, any $0 \leq t_0 \leq N$ $t_0 \leq t \leq N$, any $Q_t \in \mathcal{Q}, R_t \in \mathcal{R}, Q_N \in \mathcal{P}$, and for any $x_{t_0}$, the system is exponentially stable, i.e.*

$$
\|\Phi(t, t_0)\| \leq c_7 c_2^{t-t_0} \tag{25}
$$

*where $c_7 = \sqrt{\frac{v_{max}(\bar{P})}{v_{min}(\underline{P})}}$, $c_2 = \sqrt{1 - \frac{\mu_f}{v_{max}(\bar{P})}} \in [0, 1)$.*

*Proof.* We prove the exponential stability by constructing a Lyapunov function: $L(t, x_t) = x_t^\top P_t x_t$ for $t \geq 0$.

**Claim:** *For any $x_t$, the Lyapunov function satisfies*

$$
v_{min}(\underline{P})\|x_t\|_2^2 \leq L(t, x_t) \leq v_{max}(\bar{P})\|x_t\|^2, \quad L(t+1, D_t x_t) - L(t, x_t) \leq -\mu_f \|x_t\|_2^2.
$$

*where $v_{max}(\cdot)$ and $v_{min}(\cdot)$ denote the maximum and minimum eigenvalues of a matrix respectively.*

**Proof of Claim:** By Lemma 13, $P_t \in \mathcal{P}$, so $v_{min}(\underline{P})I_n \leq \underline{P} \leq P_t \leq \bar{P} \leq v_{max}(\bar{P})I_n$. Thus, for any $x_t$,

$$
v_{min}(\underline{P})\|x_t\|_2^2 \leq L(t, x_t) = x_t^\top P_t x_t \leq v_{max}(\bar{P})\|x_t\|^2
$$

Besides,

$$L(t+1, D_t x_t) - L(t, x_t) = x_t^\top D_t^\top P_{t+1} D_t x_t - x_t^\top P_t x_t$$
$$= x_t^\top (D_t^\top P_{t+1} D_t - P_t) x_t$$
$$= x_t^\top (-Q_t - K_t^\top R_t K_t) x_t$$
$$\leq -x_t^\top \underline{Q} x_t = -\mu_f \|x_t\|^2$$

where the third equality is by Theorem 4, the first inequality and the last equality are by $Q_t + K_t^\top R_t K_t \geq Q_t \geq \underline{Q} = \mu_f I_n$.  $\square$

By the claim above,

$$L(t+1, x_{t+1}) - L(t, x_t) \leq -\mu_f \|x_t\|_2^2 \leq -\frac{\mu_f}{v_{max}(\bar{P})} L(t, x_t)$$

Thus, $L(t+1, x_{t+1}) \leq c_2^2 L(t, x_t)$ where $c_2 = \sqrt{1 - \frac{\mu_f}{v_{max}(\bar{P})}}$. Here, $c_2$ is well-defined because $0 \leq \mu_f I_n \leq \bar{Q} \leq \bar{P} \leq v_{max}(\bar{P})$.

For any $t_{t_0}$ and any $x_{t_0}$, it is easy to verify that the state $x_t$ satisfies $x_t = \Phi(t, t_0) x_{t_0}$. Therefore,

$$\|\Phi(t, t_0)\| = \max_{x_{t_0} \neq 0} \frac{\|x_t\|}{\|x_{t_0}\|}$$
$$\leq \max_{x_{t_0} \neq 0} \sqrt{\frac{v_{max}(\bar{P})}{v_{min}(\underline{P})} \frac{L(t, x_t)}{L(t_0, x_{t_0})}}$$
$$\leq \sqrt{\frac{v_{max}(\bar{P})}{v_{min}(\underline{P})}} c_2^{t-t_0}$$

$\square$

*Step 2: show that $BK_t^\alpha \alpha_{t+1}$ is bounded.* We will show that

$$\|\alpha_t\| \leq \frac{c_7}{1 - c_2} v_{max}(\bar{P}) \bar{\theta} =: \bar{\alpha}, \qquad \|BK_t^\alpha \alpha_t\|_2 \leq \|B\|_2^2 \frac{\bar{\alpha}}{\mu_g}. \tag{26}$$

By Theorem 4, $\alpha_t$ satisfies the dynamical system $\alpha_t = D_t^\top \alpha_{t+1} + Q_t \theta_t$, with initial condition $\alpha_N = Q_N \theta_N$. As a result, we can write $\alpha_t$ in terms of $\theta_s$ and the transition matrix $\Phi(t, s)$ as follows

$$\alpha_t = Q_t \theta_t + D_t^\top Q_{t+1} \theta_{t+1} + \cdots + D_t^\top \ldots D_{N-1}^\top Q_N \theta_N$$
$$= \Phi(t, t)^\top Q_t \theta_t + \Phi(t+1, t)^\top Q_{t+1} \theta_{t+1} + \cdots + \Phi(N, t)^\top Q_N \theta_N$$

Then, the bound of $\alpha_t$ can be derived as follows.

$$\|\alpha_t\| \leq \|\Phi(t, t)^\top\| \|Q_t \theta_t\| + \cdots + \|\Phi(N, t)^\top\| \|Q_N \theta_N\|$$
$$= \|\Phi(t, t)\| \|Q_t \theta_t\| + \cdots + \|\Phi(N, t)\| \|Q_N \theta_N\|$$
$$\leq c_7 v_{max}(\bar{P}) \bar{\theta} + \cdots + c_7 c_2^{N-t} v_{max}(\bar{P}) \bar{\theta}$$
$$\leq c_7 v_{max}(\bar{P}) \bar{\theta} \frac{1}{1 - c_2} = \bar{\alpha}$$

where the second inequality is by Lemma 14, $Q_t \leq \bar{Q} \leq \bar{P}$ and $Q_N \in \mathcal{P}$.

Consequently,

$$\|BK_t^\alpha \alpha_t\| = \|B(R_t + B^\top P_{t+1} B)^{-1} B^\top \alpha_t\|$$
$$\leq \|B\|^2 \|(R_t + B^\top P_{t+1} B)^{-1}\| \|\alpha_t\|$$
$$\leq \|B\|^2 \frac{\bar{\alpha}}{\mu_g}$$

*Step 3: bound $x_t^*$.*

$$\|x_t^*\| = \|\Phi(t, t) BK_{t-1}^\alpha \alpha_t + \Phi(t, t-1) BK_{t-2}^\alpha \alpha_{t-1} + \ldots \Phi(t, 1) BK_0^\alpha \alpha_1\|$$
$$\leq c_7 \|B\|^2 \frac{\bar{\alpha}}{\mu_g} (1 + c_2 + c_2^2 + \ldots) = c_7 \frac{1}{1 - c_2} \|B\|^2 \frac{\bar{\alpha}}{\mu_g} =: \bar{x}$$

### G.5 Proof of Lemma 8

By Theorem 4 and (26), $\|\beta_k\| = \|P_k^{-1}\alpha_k\| \leq \frac{1}{v_{min}(\underline{P})}\bar{\alpha}$ for any $k$, any $N$ and any $Q_t \in \mathcal{Q}, R_t \in \mathcal{R}, Q_N \in \mathcal{P}$. When $Q_t = Q \in \mathcal{Q}, R_t = R = \mathcal{R}$ for all $t$, $\beta_k \to \beta^e$ as $k \to -\infty$ by the proof (Part i)) of Proposition 1. Thus, $\|\beta^e\| \leq \frac{1}{v_{min}(\underline{P})}\bar{\alpha}$. Define $\bar{\beta} = \max(\bar{\theta}, \frac{1}{v_{min}(\underline{P})}\bar{\alpha})$. This completes the proof.

## H  Simulation descriptions

### H.1  LQT

The experiment settings are as follows. Let $A = [0, 1; -1/, 5/6], B = [0; 1], N = 30$. Consider diagonal $Q_t, R_t$ with diagonal entries i.i.d. from $\text{Unif}[1, 2]$. Let $\theta_t$ i.i.d. from $\text{Unif}[-10, 10]$. The stepsizes of RHGD and RHTM are based on the conditions in Theorem 1. The stepsizes of RHAG can be viewed as RHTM with $\delta_c = 1/l_c, \delta_y = \delta_\omega = \frac{\sqrt{\zeta}-1}{\sqrt{\zeta}+1}$ and $\delta_z = 0$.

### H.2  Robotics tracking

Consider the following discrete-time counterpart of the kinematic model

$$x_{t+1} = x_t + \Delta t \cdot \cos \delta_t \cdot v_t$$
$$y_{t+1} = y_t + \Delta t \cdot \sin \delta_t \cdot v_t$$
$$\delta_{t+1} = \delta_t + \Delta t \cdot w_t$$

Thus we have

$$\delta_t = \arctan(\frac{y_{t+1} - y_t}{x_{t+1} - x_t})$$
$$v_t = \frac{1}{\Delta t} \cdot \sqrt{(x_{t+1} - x_t)^2 + (y_{t+1} - y_t)^2}$$
$$w_t = \frac{\delta_{t+1} - \delta_t}{\Delta t} = \frac{1}{\Delta t} \cdot \left[\arctan(\frac{y_{t+2} - y_{t+1}}{x_{t+2} - x_{t+1}}) - \arctan(\frac{y_{t+1} - y_t}{x_{t+1} - x_t})\right]$$

So that $(\delta_t, v_t, w_t)$ can be expressed by the state variables $(x_t, y_t)$.

In the simulation, the given reference trajectory is

$$x_t^r = 16\sin^3(t - 6)$$
$$y_t^r = 13\cos(t) - 5\cos(2t - 12) - 2\cos(3t - 18) - \cos(4t - 24)$$

As for the objective function, we set the cost coefficients as

$$c_t = \begin{cases} 0, & t = 0 \\ 1, & \text{otherwise} \end{cases} \qquad c_t^v = \begin{cases} 0, & t = N \\ 15\Delta t^2, & \text{otherwise} \end{cases} \qquad c_t^w = \begin{cases} 0, & t = N \\ 15\Delta t^2, & \text{otherwise} \end{cases}$$

The discrete-time resolution for online control is $0.025$ second, i.e., $\Delta t = 0.025s$. When implementing each control decision, a much smaller time resolution of $0.001s$ is used to simulate the real motion dynamics of the robot.