[Reviews · NeurIPS 2019]

Reviewer 1



SETTING: In continuous control, model predictive control is often used as a means to negotiate model mismatch or misspecification. Here, the setting assumes the availability of cost functions for some time steps in the future. This seems somewhat stylized. In terms of predictability of cost functions (representing prices etc.), a guarantee akin to that of doubly robust estimators would seem more apt. Furthermore, while regret is a perfectly well-defined quantity, a competitive ration like guarantee might be a better measure since the regret is non-vanishing. TECHNIQUES: The key observation here is that with the foreknowledge of cost functions up to a certain horizon enables one to run gradient-based offline optimization algorithms (like GD, NAG etc.) on the best-in-hindsight problem for a few iterations in the online setting. This was observed in previous work (Li 2018). This paper proposes a somewhat classical, yet very well placed reparameterization of the problem that fits into the aforementioned framework. To the reviewer, it seems the obvious parameterization in terms of the controls does not seem to, at least immediately, yield the result. **In this light, this reparameterization + offline-algorithm-on-hindsight-problem observation is quite neat.** The lower bound seems conceivable given the one in (Li 2018). 1. Can the authors comment on the transformation to the canonical form affect the strong convexity and smoothness of the transformed costs on Line 479? 2. Line 155/6 -- Was x_t+1 = A x_t + B u_t intended instead of x_t+1 = x_t + u_t? POST RESPONSE Adding the response under "Parameterization in terms of control inputs." to the main paper would be nice. The reviewer also thinks R#3's suggestion of putting GD first would add to readability. The reviewer is retaining his/her score.

Reviewer 2



The paper and the contributions are cleanly written. Matching upper/lower bounds on regret and proposed algorithm would be significant to community. One issue is upper/lower bounds still differ by condition number^2 as discussed below. The fact that setup applies to possibly nonlinear f_t,g_t is definitely good although it is not clear how much technicality is introduced by doing this compared to LQT setup. -------------------------------------------------------- After reading other reviews and author feedback, I am more convinced about the significance of the paper and adjusted my score accordingly.

Reviewer 3



The authors study online control problems where the dynamics are known, there is no process noise, but the cost functions are changing over time. A main motivating application is LQ tracking. The authors study the problem by looking at dynamic regret, where the online algorithm has to compete with the optimal solution (not just best static controller in hindsight). To compensate for this more difficult objective, the algorithm is assumed to have access to a small prediction lookahead window. The main result of the work is a new momentum based online control algorithm. The authors prove that the regret of their proposed algorithm decreases exponentially in the prediction lookahead window size W. The authors also prove a lower bound that shows that for LQ tracking their regret upper bound is nearly optimal. Experimentally, the authors show their algorithm outperforms MPC from a regret point of view while using similar amount of computation. This is a nice paper. The focus on dynamic regret is new, and the reduction from LQ controllable form to unconstrained convex optimization is of interest to the online learning community. The authors should cite the following related paper: Online Control with Adversarial Disturbances. Agarwal et al. ICML 2019. There are some similarities between Agarwal et al. and this work. For instance, Agarwal et al. also considers time varying costs while allowing for an adversarial disturbance sequence to affect the dynamics. However, they show regret bounds with respect to the best (almost) static policy. Presentation wise, in Algorithm 1 I would have preferred to first see the algorithm with vanilla gradient descent first before moving to the triple momentum algorithm. The regret bound can be presented for just RHTM, but for simply understanding the algorithm a simpler update rule would be easier to follow. Minor typo: in lines 155 and 156, it should be x_{t+1} = A x_t + B u_t.

[Author Response · NeurIPS 2019]

We are grateful to hear that the reviewers find our manuscript worth of being accepted. All reviewers make good
summaries of our contributions and provide helpful suggestions for us to further improve the paper. Here we provide
some specific discussions to address the reviewers' questions. We will also incorporate them into our final version.

==== *To reviewer 1* ====

**Prediction model.** We agree with the reviewer that our current prediction model is idealized and should be generalized.
Indeed, our ongoing work is considering noisy predictions and studying the role of bias and variance of the prediction.
One issue we face is finding a meaningful yet simple enough noisy prediction model for insightful theoretical analysis.
We really appreciate the suggestion of the doubly robust estimators and will try it in our ongoing work.

**Competitive ratio.** This is a good point and in fact we had investigated whether a constant competitive ratio could be
obtained before our initial submission. However, we have not obtained many meaningful results without adding strong
assumptions on the objectives. We will continue to work on this as future work and discuss it in the final version.

**Parameterization in terms of control inputs.** We agree that the more direct parameterization in terms of controls
does not yield the result. [Richter, Jones, Morari, 2012] shows that when the system matrix $A$ is unstable, the condition
number of the $W$-step optimization on the control inputs goes to infinity as $W \to +\infty$, causing numerical issues. But
for our parameterization, the condition number is independent of $W$ and is bounded even when $A$ is unstable.

**Effect of linear transform to canonical form on the strong convexity and smoothness.** This is a good point. The
strong convexity and smoothness are preserved under the linear transformation to the canonical form, but the strong
convexity number and the smoothness number would change with the transformation. Taking $\hat{f}_t(\hat{x}_t) = f_t(S_x^{-1}\hat{x}_t)$ as
an example, the smoothness of $\hat{f}_t$ can be upper bounded by $\|S_x^{-1}\|_2^2 l_f$, and the strong convexity is lower bounded by
$\mu_f/\|S_x\|^2$. Roughly speaking, not only $f_t, g_t$ but also $A, B$ affect the condition number $\zeta$. Getting more elegant and
insightful formulas are worth exploring and left as future work. More discussion will be added to the paper.

**On the** $x_{t+1} = x_t + u_t$**.** Yes, this is a typo and should be $x_{t+1} = Ax_t + Bu_t$.

==== *To reviewer 2* ====

**Infinite regret when** $\zeta \to +\infty$**?** The regret bound will not go to infinity as $\zeta \to +\infty$ because $((\sqrt{\zeta}-1)/\sqrt{\zeta})^K \leq 1$
$((\sqrt{\zeta}-1)/(\sqrt{\zeta}+1))^K \leq 1, \forall \zeta \geq 1$. When $\zeta$ is large, the regret decays slowly when prediction $W$ increases.

**The strong convexity of** $g_t$ **is not used.** This is true and is a good observation. We will clarify this in the final version.

**The gap between the upper and lower bounds.** We think it would be extremely exciting, but not an easy task,
to close this gap, because this gap is closely related to a long lasting gap between the upper and lower bounds of the
convergence rate of the first-order optimization algorithms for strongly convex and smooth functions. In particular, our
regret upper bound is based on the convergence rate of triple momentum (the best convergence rate to our knowledge)
[39], and our regret lower bound, perhaps surprisingly, matches the convergence rate lower bound by Nesterov [38].
This interesting connection is worth further exploring. Research advances for either gap will help close the other gap.
Here, we leave this gap as future work since our major focus, as summarized by the reviewer, is to quantify the effect
of prediction in online control instead of solving the long lasting open question in first-order optimization.

**Comparing with learning-based control and noisy unknown dynamics.** In some sense, our current results are
orthogonal to that of learning-based control, because learning-based control usually considers a time-invariant en-
vironment and aims to learn system parameters or optimal controllers by data; while our current paper considers a
time-varying scenario with known dynamics but changing objectives and studies decision making with limited predic-
tions. We agree that assuming noiseless known dynamics is a main drawback of our setup. It is our ongoing work to
relax the assumptions to generalize the application of our results. Firstly, we note that our current setup can handle
deterministic but time-varying disturbances in the same way of handling the time-varying objectives. However, for
general randomness and unknown dynamics, it requires much more work. One way is to draw ideas from learning-
based control, trying both model-based and model-free learning. To get meaningful results, we conjecture that the
environment's volatility and nonasymptotic learning guarantees will be crucial. We also hope certainty equivalence
will facilitate the regret analysis to some degree. More references and discussions will be added to the paper.

**Why nonlinear** $f_t, g_t$**?** Our algorithm naturally applies to general $f_t, g_t$ and the proof only uses strong convexity and
smoothness. But we agree LQR is a nice candidate and allows more customized results as shown in section 5.

==== *To reviewer 3* ====

**Comparison to [Agarwal et al. 2019].** Thank you very much for mentioning this. We also found this paper, but
shortly after our submission. It is very relevant and we will cite it and compare with it in the final version.

**Algorithm presentation.** We thank the suggestion and will present the vanilla gradient method first in the final paper.

[Meta-Review · NeurIPS 2019]

Congratulations on your work. Please do not forget to make the necessary modifications to your paper for the final version.